SOFTWARE

# NRV: An open framework for *in silico* evaluation of peripheral nerve electrical stimulation strategies

**Thomas Couppey[1]**, **Louis Regnacq[1,2]**, **Roland Giraud[1,2]**, **Olivier Romain[1]**, **Yannick Bornat[2]**, **Florian Kolbl[1,2]** *

**1** Laboratoire ETIS, Cergy Paris Université, ENSEA, CNRS UMR 8051, Cergy, France, **2** Univ. Bordeaux, CNRS, Bordeaux INP, IMS, UMR 5218, Talence, France

☯ These authors contributed equally to this work.
* florian.kolbl@ims-bordeaux.fr

**Data Availability Statement:** Data for reproducing results, appart from supplementary material sent with the submission are linked to https://doi.org/10.5281/zenodo.10497741.

## Abstract

Electrical stimulation of peripheral nerves has been used in various pathological contexts for rehabilitation purposes or to alleviate the symptoms of neuropathologies, thus improving the overall quality of life of patients. However, the development of novel therapeutic strategies is still a challenging issue requiring extensive *in vivo* experimental campaigns and technical development. To facilitate the design of new stimulation strategies, we provide a fully open source and self-contained software framework for the *in silico* evaluation of peripheral nerve electrical stimulation. Our modeling approach, developed in the popular and well-established Python language, uses an object-oriented paradigm to map the physiological and electrical context. The framework is designed to facilitate multi-scale analysis, from single fiber stimulation to whole multifascicular nerves. It also allows the simulation of complex strategies such as multiple electrode combinations and waveforms ranging from conventional biphasic pulses to more complex modulated kHz stimuli. In addition, we provide automated support for stimulation strategy optimization and handle the computational backend transparently to the user. Our framework has been extensively tested and validated with several existing results in the literature.

## Author summary

Electrical stimulation of the peripheral nervous system is a powerful therapeutic approach for treating and alleviating patients suffering from a large variety of disorders, including loss of motor control or loss of sensation. Electrical stimulation works by connecting the neural target to a neurostimulator through an electrode that delivers a stimulus to modulate the electrical activity of the targeted nerve fiber population. Therapeutic efficacy is directly influenced by electrode design, placement, and stimulus parameters. Computational modeling approaches have proven to be an effective way to select the appropriate stimulation parameters. Such an approach is, however, poorly accessible to inexperienced users as it typically requires the use of multiple commercial software and/or development

**Funding:** This research is part of the BioTIFS (Improved Selectivity for Bioelectronic Therapies with Intrafascicular Stimulation) project, and is supported by the Collaborative Research in Computational Neuroscience (CRCNS) program, by the French Agence Nationale pour la Recherche (ANR-18-NEUC0002-02) and by the U.S. National Institutes of Health (NIH-R01-EB027584). The funder had no role in study design, data collection and analysis, decision to publish, or preparation of the manuscript.

**Competing interests:** The authors have declared that no competing interests exist.

in different programming languages. Here, we describe a Python-based framework that aims to provide an open-source turnkey solution to any end user. The framework we developed is based on open-source packages that are fully encapsulated, thus transparent to the end-user. The framework is also being developed to enable simulation of granular complexity, from rapid first-order simulation to the evaluation of complex stimulation scenarios requiring a deeper understanding of the ins and outs of the framework.

## Introduction

The network of peripheral nerves offers extraordinary potential for modulating and monitoring the functioning of internal organs or the brain. Recent developments in electroceuticals have shown promising results, highly targeted treatment, and fewer side effects compared to conventional drug-based therapies [1–5]. Electrical stimulation works by delivering an electrical current through specialized electrodes to a targeted neural tissue in the nervous system to induce or modulate desired neural activity [6]. Currently, electrical stimulation is used in several applications in the central nervous system (CNS), such as deep brain stimulation (DBS) for Parkinson's disease [7] or for epileptic seizure reduction [8], and is also being studied in the peripheral nervous system (PNS) for applications such as motor rehabilitation [9], restoration of some involuntary or visceral functions [10] for example. Recently, non-conventional stimulation delivering a specific kilohertz continuous electrical waveform to the neural target has been used to block unwanted neural activity for pain relief [11].

Although the literature reports many encouraging novel bioelectronic therapies, many of them fail to reach a higher level of technology readiness and don't succeed in translating to the market [12–17]. Indeed, the sizeable morphological variety across species and within individuals of the same species greatly impacts the electrical stimulation therapy outcomes [18–20]. Also, the electrode design and its spatial location and orientation, as well as the choice of the electrical stimulus will further increase the lack of predictability of the therapy [21–23]. Computational modeling has proven to be a valuable tool to tackle those issues but also for accelerating the design of the therapy while containing the financial cost of the *in vivo* experiments [24–28]. In addition, the use of in *in silico* studies facilitates the adoption of the "Four Rs" ethical guidelines (Reduction, Refinement, Replacement, and Responsibility) required by *in vivo* experiments [29]. Ultimately, *in silico* models provide access to individual fiber responses and insight into the internal states of the neurons that constitute the simulated neural target. It helps to better understand the complex behavior of neural fibers and enables the design of specific therapies [30–32].

The state-of-the-art process adopted by the community for simulating the PNS response to electrical stimulation is based on a two-step hybrid model: [19, 33–38]:

1. A realistic 3D model is created. It includes the anatomical features as well as an accurate representation of the electrode design and location in or around the neural tissue. A finite element modeling (FEM) solver is commonly used to compute the spatial extracellular electric potential distribution in the tissue induced by the electrical stimulation. This step usually relies on the quasi-static approximation [39].

2. The extracellular potential is then applied to unidimensional compartmental models that simulate the response of axon fibers to the electrical stimuli. In the case of PNS fibers, the ephaptic coupling between fibers is usually neglected [34, 40, 41] or taken into account for very specific studies [42]. Hodgkin-Huxley-like (i.e. nonlinear conductance-based) models

are commonly used, and distinct models are developed for specific trans-membrane protein and associated types of fiber (myelinated [43], unmyelinated [44], afferent/efferent [45], etc.).

In this paper, we propose and describe a fully open-source PNS simulation framework, free of any commercial license. The framework is based on the Python language, is multi-platform (available on Windows, OS-X, and Linux), and can be effortlessly deployed on a conventional machine (e.g., a laptop) or a supercomputer without changing a single line of code. This framework has been designed as an abstraction layer of the physical, mathematical, and computational techniques required to realistically predict and study PNS electrical stimulation. This framework has also been designed to be linked to optimization algorithms and experimental setups to facilitate translation between novel stimulation protocol development and experimental campaigns.

The methods section begins with a literature review of existing solutions and then describes the architecture of our framework, the algorithms, the physics, and the hypotheses behind the simulation with references to the existing literature. In the results section, we validate the output of the framework with existing results and show its potential applications. Finally, we discuss the advantages of our approach over the related literature and future development directions.

## Design and implementation

### Overview of existing solutions

An extensive description of the processes for obtaining and simulating hybrid models was first introduced by Raspopovic *et al.* and then refined by Romeni *et al.* [33, 34]. The authors provided a systematic methodology to build and exploit realistic hybrid models for designing PNS interfaces. The workflow was demonstrated using COMSOL Multiphysics (COMSOL AB, Stockholm, Sweden) and MATLAB (The MathWorks Inc, Massachusetts, United States) to build the geometry and solve the FEM. The neural dynamic was computed using NEURON [46]. Some algorithms and examples are made available on a public repository. However, the authors of [33, 34] did not provide a ready-to-use solution, making the deployment of the workflow challenging. Recently, however, several freely available solutions based on the same or similar methodology have been developed and introduced to the community to facilitate the use of hybrid modeling. These solutions are presented in this section.

**PyPNS.**  PyPNS is the first open-source, Python-based framework for electrical stimulation and extracellular recording of a nerve of the PNS that integrates axon models and extracellular potential in a single environment [40]. However, the extracellular potential must be pre-calculated by an external FEM solver. In their original paper, Lubba et al. used COMSOL to solve the extracellular electric field, which requires a commercial license. Neural dynamics are solved using the NEURON simulation software via the Python API [47]. In PyPNS, the nerve geometry is considered as a cylindrical shape filled with epineurium and surrounded by saline. Stimulation and recording are performed with a cuff-like electrode, and arbitrary waveform stimuli can be used. The MRG model [43] is used to simulate myelinated fibers and the Sundt model [48] for unmyelinated fibers. The PyPNS framework also aims to improve the simulation by taking into account the tortuosity of the axons in the PNS nerve.

**ASCENT.**  ASCENT [49] provides a computational modeling pipeline for simulating the PNS that closely follows the methodology proposed in [34]. The pipeline uses the Python and Java languages, COMSOL Multiphysics for the FEM problem and NEURON for solving the neural dynamics. The simulation parameters are defined in JSON configuration files, and the

simulation is a two-step process: i) Python, Java, and the commercially licensed COMSOL Multiphysics software are used to evaluate the potentials along the neural fibers. ii) Python and NEURON are used to compute the resulting neural activity. The second step can easily be performed on a computer cluster. A standardized pipeline was developed to describe and process nerve morphology, which facilitates the creation and simulation of histology-based or highly realistic *ex novo* nerve geometry. ASCENT provides several templates for cuff electrodes based on commercially available parts, which can be further customized. Arbitrary waveforms can be generated to simulate neural activation and block. The MRG model [43] is used to simulate myelinated fibers and the Rattay and Aberham, Sundt et al. and Tigerholm et al. models [44, 48, 50] are available for unmyelinated fibers. The capabilities for simulated extracellular recordings using a computationally efficient and accurate filtering method have been implemented since a recent update of the pipeline (ASCENT 1.3.0) [51]. The pipeline also provides tools for analysis and visualization such as heatmaps of thresholds or video generators for plotting state variables variation over time and space. ASCENT is open-source, freely available on a repository, and extensive examples and documentation materials are provided.

**TxBDC Cuff Lab.** The TxBDC Cuff Lab [52] approach is based on a Python web server with an online front-end graphical user interface (GUI) that can run simple PNS electrical stimulation models. In TxBDC, nerves consist of a cylindrical shape containing multiple ellipsoidal fascicles. The FEM model is meshed with Gmsh [53] and solved under the quasi-static assumption using the FEniCS library [54]. Both Gmsh and FEniCS are open source and don't require a commercial license. Electrodes in TxBDC can be either parameterized cuff-like or intraneural electrodes. The stimulation waveform is limited to either a monophasic or a biphasic pulse. Neural dynamics are calculated using passive axon models. The passive model is based on 3-D fitted curves obtained from a pre-calculated active axon model modeled with NEURON. A total of 20 fitted curves are pre-calculated and used to estimate neural dynamics. Overall, TxBDC is very simple and intuitive as the simulation is entirely parameterized via the GUI (available online). The TxBDC backend Python sources are freely available in a dedicated repository.

**ViNERS.** ViNERS [41] is an open-source MATLAB-based toolbox that focuses on the modeling of visceral nerves. Specifically, the FEM is handled within the toolbox and is based on Gmsh for meshing the geometry and is solved with EIDORS [55]. Both Gmsh and EIDORS are open source. The neural dynamics are solved using NEURON. Nerve anatomy can be imported from a traced nerve section of a histology specimen or an *ex novo* anatomy can be created. ViNERS implements parameterized cuff and planar electrode arrays. User-defined electrodes can be added. Nerve design and electrode configuration can be done via scripting or with a JSON configuration file. The toolbox provides a GUI to generate the JSON. ViNERS also supports custom arbitrary waveforms. Unmyelinated/myelinated axons are populated from the histology-based fiber distribution. The MRG and Gaines [45] models are used for myelinated fibers and the Sundt model is used for unmyelinated fibers. FEM-based extracellular recording of neural activity can also be simulated with the toolbox. ViNERS provides tools to analyze the simulation output such as spiking threshold detection or single fiber action potential detection for recordings. ViNERS is provided as a freely downloadable MATLAB toolbox. Source code and examples are also provided.

The development of dedicated solutions for hybrid modeling is a major step forward towards simple, efficient and easily reproducible *in silico* electrical stimulation of the PNS. However, with the exception of the TxBDC Cuff Lab approach, none of the presented solutions are fully based on open-source and free-to-use solutions, which limits the ability to replicate experiments and reuse data [56]. ASCENT and ViNERS are open-source but require closed-source and commercially licensed software to run: ViNERS is based on MATLAB, and

ASCENT requires COMSOL Multiphysics to run. PyPNS is also open-source but does not include the FEM solver, and the demonstration used COMSOL Multiphysics to precalculate the FEM. As for the TxBDC Cuff Lab approach, it is fully open-source and free to use, but too restrictive as it is limited to simplified nerves, electrode geometries, and stimulation waveforms. Alternatively, a variety of open-source and non-commercially licensed software and libraries are available to perform some aspects of the hybrid modeling process, but the entire process requires the use and knowledge of multiple software and/or programming languages. This results in a high cost of entry for inexperienced users and makes it complex to deploy hybrid model simulations on clusters of supercomputers or massively parallel machines.

## Our approach: NRV overview

The NeuRon Virtualizer framework (NRV) is a fully open-source multi-scale and multi-domain Python-based framework developed for the hybrid modeling of electrical stimulation in the PNS. NRV shares the same working principle as the solutions presented in the previous section: a FEM model is solved under the quasi-static assumption to compute the extracellular potential generated by a source and is applied to a neural model to estimate the resulting neural activity. However, NRV improves on the other solution by being able to perform all the necessary steps for hybrid modeling within the framework, and without the need for commercially licensed software or library. Realistic electrode geometries and parameterized cylindrical nerves and fascicles are constructed and meshed using Gmsh [53], and FEM problems are solved using the FEniCS software [54]. An optional bridge between COMSOL Multiphysics and NRV via COMSOL Server (requiring a commercial license) is also implemented in NRV providing multiple options to the users for creating the 3-D model.

Tools for generating realistic axon population and placement are provided and commonly used in the literature axon computational models are implemented. Extracellular potential computed with the FEM model is interpolated and used as input for the 1-D axon models. An intracellular voltage or current clamp can also be used to stimulate one or multiple axons of an NRV model. Axon models are simulated using the NEURON software via the NEURON to Python bridge. All computation inputs and outputs are stored in dictionary objects to enable context saving and facilitate data reuse. Post-processing tools are also provided to automatically detect action potentials, filter the data, etc. Extracellular recorders are also implemented and can be added to the model to simulate electrically-evoked compound action potentials (eCAPs).

Optimization algorithms have been successfully applied to improve electrode design [57] or stimulation waveforms or protocol [58–60]. However, this technique requires an important phase of formalization and software development. NRV extends other presented solutions by also including a formalism, methods, and algorithms to provide an out-of-the-box solution for describing any optimization problem for PNS simulation within the framework.

Regarding computational performances, NRV provides multiprocessing capabilities by implementing a message-passing interface (MPI) for Python [61] to speed up the computation of axon dynamics through parallel invocations of the NEURON solver. Parallel computing is performed independently from the simulation description and the end-user only needs to provide the maximum number of usable cores on the machine; NRV automatically handles job distribution and synchronicity between the processes.

Calls to Gmsh, FEniCS, and COMSOL Server (for geometry definition, meshing, FEM solving, and electric field dispatching), NEURON (for neural dynamic solving), or other third-party libraries used (for simulation post-processing for example) are fully encapsulated in the NRV framework. Interactions with these libraries are not required to use the framework and

therefore do not require any special understanding of the end user's point of view. NRV aims to be accessible to users with only basic Python experience, as well as easily readable from a high-level simulation perspective. NRV also enables multi-scale simulations: single axonal fibers to whole nerve simulations can be performed with NRV and require only a couple of lines of code. The framework is pip installable making the framework effortlessly deployable on a computer cluster or supercomputer.

NRV's internal architecture is depicted in Fig 1. It is subdivided into four main sections:

- `fmod`: handles 3-D extracellular model generation and computation.

- `nmod`: manages 1-D axon membrane potential model description and computation.

- `optim`: enables automated optimization of stimulation contexts, either by controlling geometrical parameters or the stimulation waveshape for instance.

- `backend`: manages all related software engineering aspects behind NRV, such as machine capacity and performance, parallel processing, or file input and output. This section is not related to the neuroscientific computational aspect and is not described further below.

## fmod-section: Computation of 3D extracellular potentials

NRV provides classes, tools, and templates to create 3D models of the nerve and electrodes in the `fmod`-section of the framework. Fig 2 provides a synthetic overview of the extracellular simulation problem and a simplified UML-class diagram of the software implementation, including class references accessible to the end user. In this paragraph, we provide details of the implementation of the physics and the corresponding computational mechanisms.

The estimation of the extracellular electric potential resulting from the electrical stimulation is handled by the `extracellular_context`-class. The `analytical_stimulation`-class and the `FEM_stimulation`-class are derived from the parent `extracellular_context`-class as illustrated in Fig 2 and detailed in the next two subsections. The extracellular potential generated by neural activity is handled by the `recorder`-class and described later on.

**Electrical stimulation potential: Computation mechanism.** NRV relies on the assumption of linear impedance material, so that the contribution of each electrical source, i.e. stimulating electrode, can be linearly combined. In addition, the quasi-static approximation leads to a time-decoupled solution of the electrical stimulation potential. As so, for any point of coordinate $\mathbf{r}$ in space ($\mathbf{r} \in \mathbb{R}^3$) at a time $t$, the extracellular electrical potential is computed assuming linearity as:

$$V_{ext}(\mathbf{r}, t) = \sum_{k \in \mathcal{E}} I_{\text{stim } k}(t) V_{\text{footprint } k}(\mathbf{r}) \tag{1}$$

where $\mathcal{E}$ is the set of electrodes in the simulation, $I_{\text{stim } k}$ is the stimulation current at the electrode $k$, and $V_{\text{footprint } k}(\mathbf{r})$ is a function of $\mathbb{R}^3 \to \mathbb{R}$, in $\text{V} \cdot \text{A}^{-1}$, computed once for each electrode before any simulation and which corresponds to the 3D extracellular electrical potential generated by the electrode for a unitary current contribution. The footprint function can also be physically interpreted as a transfer resistance (being homogeneous to $\Omega$) between $\mathbf{r}$ and the electrode location. This electrode contribution function can be computed using two different methods, explained below, and is referred to in this article as the electrode footprint.

The stimulation current $I_{\text{stim}, k}$ is described by a dedicated `stimulus`-class. Such object have two attributes that consist in stimulus current $\{s_1, \cdots, s_n\}$ and corresponding $\{t_1, \cdots, t_n\}$

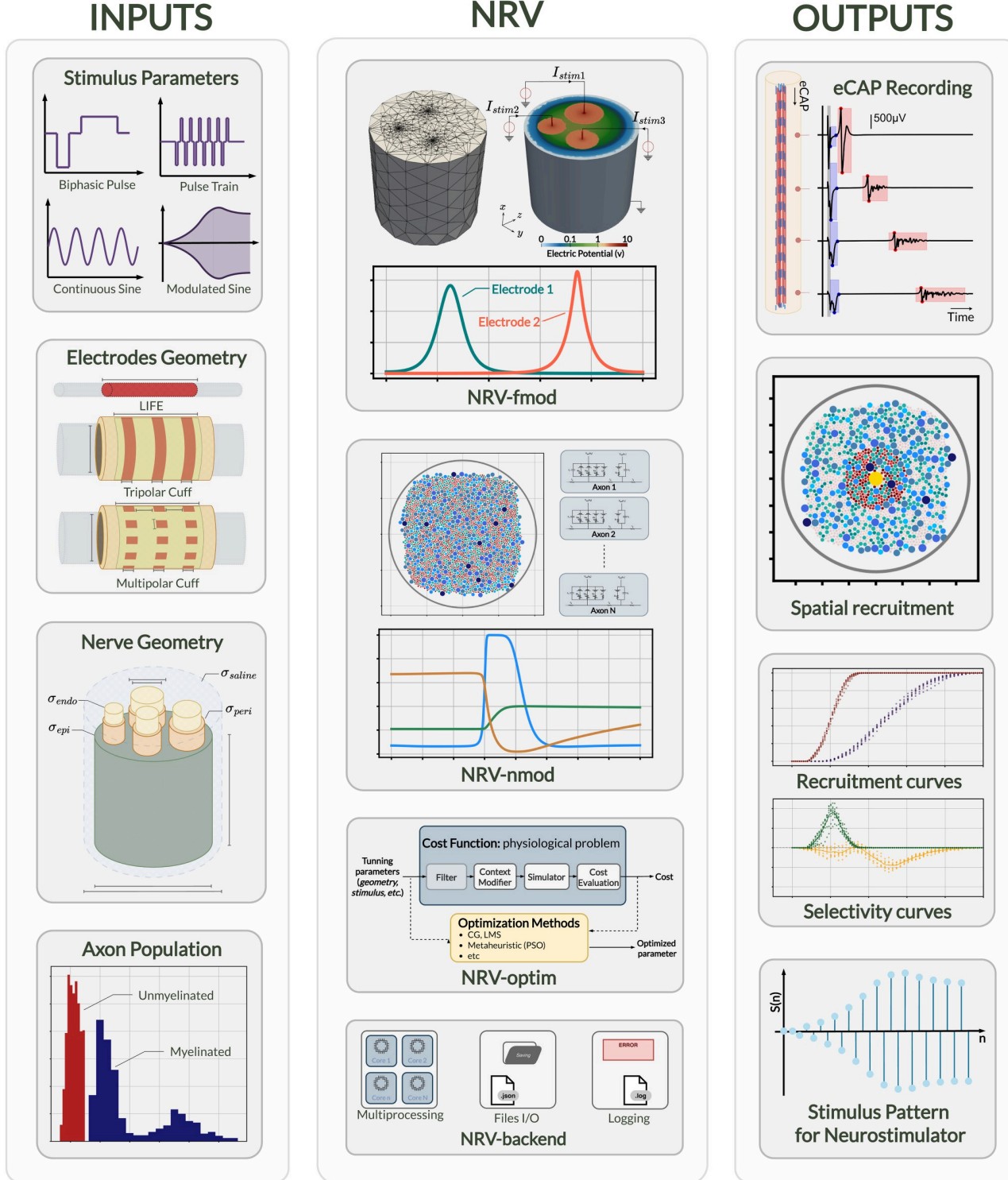

**Fig 1. Schematic overview of NRV.** The figure shows the framework inputs (stimulus parameters, electrode and nerve geometries, and axon populations), the framework mains sections (*fmod*, *nmod*, *optim*, and *backend*) described in the **Method** section of this paper, and some possible result outputs of the NRV framework.

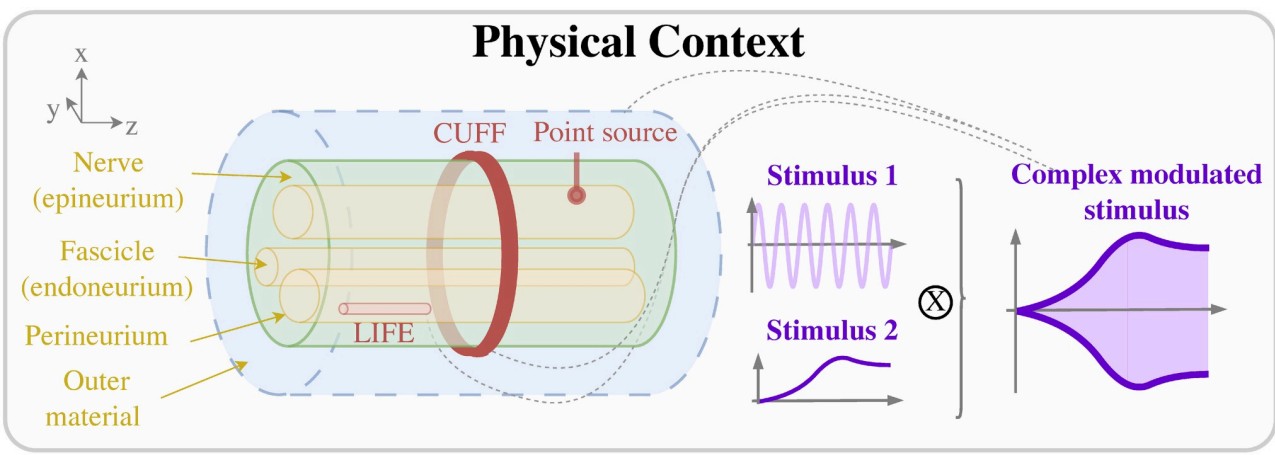

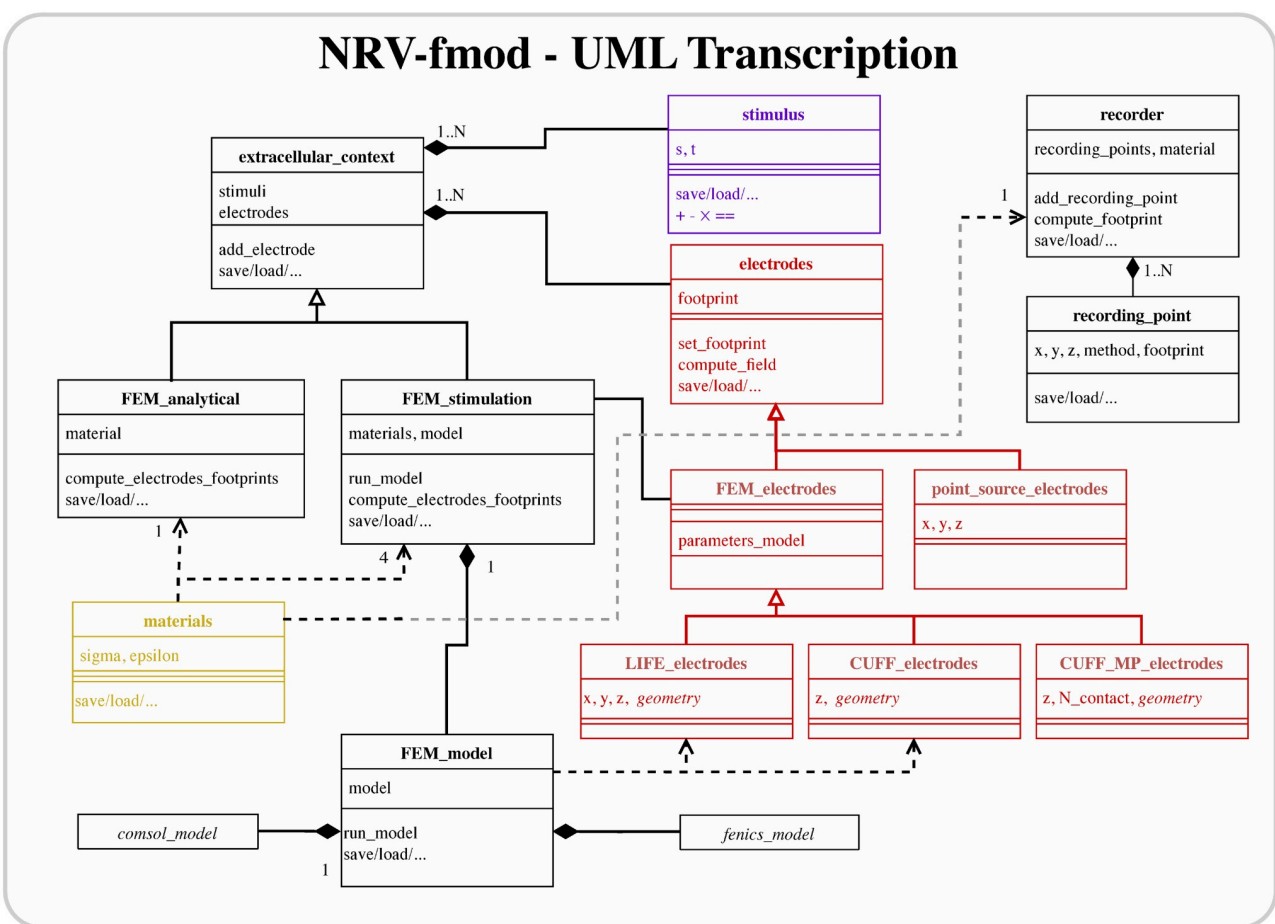

**Fig 2. Fmod section of the NRV framework.** Top: Schematic conceptualization of a PNS extracellular simulation context. The context is created around a nerve, its fascicles, and corresponding materials. Electrodes can be added as electrical interfaces, each electrode contact being associated with a simple or complex stimulus (or stimulation waveform). Bottom: NRV's transcription of the physical model with dedicated classes that can be combined and used by the framework to evaluate the extracellular electrical fields.

time lists of values applied to an electrode $k$. Stimuli are asynchronous signals, i.e. they don't need to be synchronized to a global time reference. Arithmetic and logical operations between `Stimulus`- objects are defined for the class by operator overloading, and consist in computing the resulting $\{s_1, \cdots, s_m\}$ and corresponding $\{t_1, \cdots, t_m\}$ of the operation, enabling fully customizable arbitrary stimulus waveforms (see S1 Text and S1 File for some examples). This approach facilitates the design of complex waveforms such as modulated stimuli [58, 62, 63]. NRV also includes dedicated methods for fast generation of monophasic and biphasic pulses, sine-wave, square-wave, and linear ramps stimulus.

As described in the class diagram of Fig 2, the end user can add an electrode (`electrode`-class) and stimuli (`stimulus`-class) combination to the model. Each `electrode`-object in NRV has a unique identifier (ID) and multiple electrodes can be added to the simulation model. The end user can choose between two different methods to compute the extracellular potential: an analytical approach or a FEM approach. If the extracellular potential is solved analytically, only Point-Source Approximation (PSA) electrodes can be implemented limiting the simulation to geometry-less approximation [64].

Using the FEM approach, classes to simulate cuff-like electrodes and longitudinal intrafascicular electrodes (LIFEs) [65] are implemented in the framework. FEM electrodes can be fully parameterized (active-site length, number of contacts, location, etc.). Custom classes for more complex electrode designs can be added by inheritance of the `FEM_electrodes`-class. All footprint computations are automatically performed by the `electrode`-mother class when the `extracellular_context`-object is associated with axons (mechanism described in the next section).

Electrical conductivities (isotropic or anisotropic) of the tissues constituting the NRV nerve are defined using `Material`-class. The framework includes pre-defined materials for the epineurium, endoneurium, and perineurium conductivities with values commonly found in the literature [66–70]. Material properties predefined in the framework are summarized in S1 Table. Custom conductivity values can also be user-specified for each material of the model.

**Analytical evaluation of the extracellular potential.** The `analytical_stimulation`-class solves the extracellular potential analytically using the PSA for the electrode, and the nerve is modeled as an infinite homogeneous medium [71]. This method is only suitable for geometry-less simulation: axons are considered to be surrounded by a single homogeneous material. In this case, the footprint function is computed as:

$$V_{\text{footprint}} = \frac{1}{4\pi\sigma||\mathbf{r} - \mathbf{r_e}||} \tag{2}$$

where $||\cdot||$ denote the euclidean norm, $\mathbf{r_e}$ is the $(x_e, y_e, z_e)$ position of the PSA electrode and $\sigma$ is the isotropic conductivity of the material. The conductivity of the endoneurium is generally considered as anisotropic [66] and is expressed as a diagonal matrix:

$$\boldsymbol{\sigma} = \begin{bmatrix} \sigma_{xx} & 0 & 0 \\ 0 & \sigma_{yy} & 0 \\ 0 & 0 & \sigma_{zz} \end{bmatrix} \tag{3}$$

The expression of the footprint function becomes [72]:

$$V_{\text{footprint}} = \frac{1}{4\pi\sqrt{\sigma_{yy}\sigma_{zz}(x-x_e)^2 + \sigma_{xx}\sigma_{zz}(y-y_e)^2 + \sigma_{xx}\sigma_{yy}(z-z_e)^2}} \tag{4}$$

The analytical approach provides a simple and quick estimation of the extracellular potential, allowing for fast computation on resource-constrained machines. However, it restricts the nerve geometry to an infinite homogeneous medium and omits the electrode shape and interface, limiting the viability of this approach for modeling complex experimental or therapeutic setups [73].

**FEM evaluation of the extracellular potential.** The extracellular potential evaluation in a realistic nerve and electrode model using the FEM approach is handled by the `FEM_stimulation`-class. A nerve in NRV is modeled as a perfect cylinder and is defined by its diameter, its length, and the number of fascicles inside. The position and diameter of each fascicle on the NRV nerve can be explicitly specified. Fascicles of the NRV model are modeled as bulk volumes of endoneurium surrounded by a thin layer of perineurium tissue [74]. The remaining tissue of the nerve is modeled as a homogeneous epineurium. The nerve is plunged into a cylindrical material, which is by default modeled as a saline solution.

The NRV framework offers the possibility of using either COMSOL Multiphysics or FEniCS to solve the FEM problem. For the first one, mesh and FEM problems are defined in `mph` files which can be parameterized in the `FEM_stimulation`-class to match the extracellular properties, and all physic equations are integrated into the `Electric Currents` COMSOL library. When choosing the FEniCS solver, NRV handles the mesh generation using Gmsh, the bridge with the solver, and the finite element problem with FEniCS algorithms. Physic equations solved are defined within the NRV framework. COMSOL Multiphysics is commonly used for the simulation of neural electrical stimulation investigation, but it requires a commercial license to perform computation, and all future developments are bound to the physics and features available in the software. We included the possibility of using it as a comparison to existing results but the use of FEniCS and Gmsh enables fully open-science and the possibility to enhance simulation possibilities and performances.

For both FEM solvers, the extracellular electrical potential $V_{\text{footprint}}$ in the simulation space $\Omega$, is obtained from the current conservation equation under the quasi-static approximation and the Poisson equation in a conductive material of conductivity $\sigma$:

$$\nabla \mathbf{j}(\mathbf{r}) = 0 \tag{5}$$

$$\mathbf{j}(\mathbf{r}) = \sigma(\mathbf{r})\nabla V_{\text{footprint}}(\mathbf{r}), \forall \mathbf{r} \in \Omega \tag{6}$$

where $\mathbf{j}$ is the current density. The electrical ground and the current injected by an electrode are set by respectively Dirichlet and Neumann boundary conditions:

$$V_{\text{footprint}}(\mathbf{r}) = 0, \forall \mathbf{r} \in \partial\Omega_G \tag{7}$$

$$\sigma(\mathbf{r})\nabla V_{\text{footprint}}(\mathbf{r}) \cdot \mathbf{n} = j_E(\mathbf{r}), \forall \mathbf{r} \in \partial\Omega_E \tag{8}$$

$$\sigma(\mathbf{r})\nabla V_{\text{footprint}}(\mathbf{r}) \cdot \mathbf{n} = 0, \forall \mathbf{r} \in \partial\Omega \setminus \{\partial\Omega_G, \partial\Omega_E\} \tag{9}$$

where the ground surface $\partial\Omega_G$ can be specified by the end user in one or multiple external surfaces of the geometrical model. At the electrode interface, $\mathbf{n}$ is the normal vector to the surface $\partial\Omega_E$ and the normal current density injected $j_E$ considered homogeneously distributed is

computed by:

$$\mathbf{j_E}(\mathbf{r}) \cdot \mathbf{n} = \frac{I_{stim}}{S_E}, \forall \mathbf{r} \in \partial\Omega_E \tag{10}$$

where $I_{stim}$ is the stimulation current and $S_E$ the electrode contact surface area.

To reduce the number of elements in the mesh associated with smaller material dimensions, the fascicular perineurium volumes are defined using the thin-layer approximation (see Fig 3) [74, 75]. The current flow is assumed to be continuous through the layer, while a discontinuity is induced in the potentials:

$$\mathbf{j_{lay}} = \sigma_{in}\nabla V_{in} = \sigma_{out}\nabla V_{out} \tag{11}$$

$$\mathbf{j_{lay}} \cdot \mathbf{n_{e/i}} = \frac{\sigma_{lay}}{t_h}(V_{in} - V_{out}) \tag{12}$$

**Simulation of eCAP recordings: Computation mechanism.** In NRV, eCAPs are computed analytically only, using point- or line-source approximations (PSA or LSA) [76] for the contribution of each axon in the simulation. Using the linear material impedance hypothesis, the total extracellular electrical potential can be considered as the sum of the contribution from the stimulating electrodes and the neural activity of the axon. Thus, the two contributions can be calculated separately. The geometry is only based on one material (by default endoneurium). This strategy ensures computational efficiency while still providing sufficiently quantitative results about axon synchronization and eCAP propagation for comparison with experimental observations.

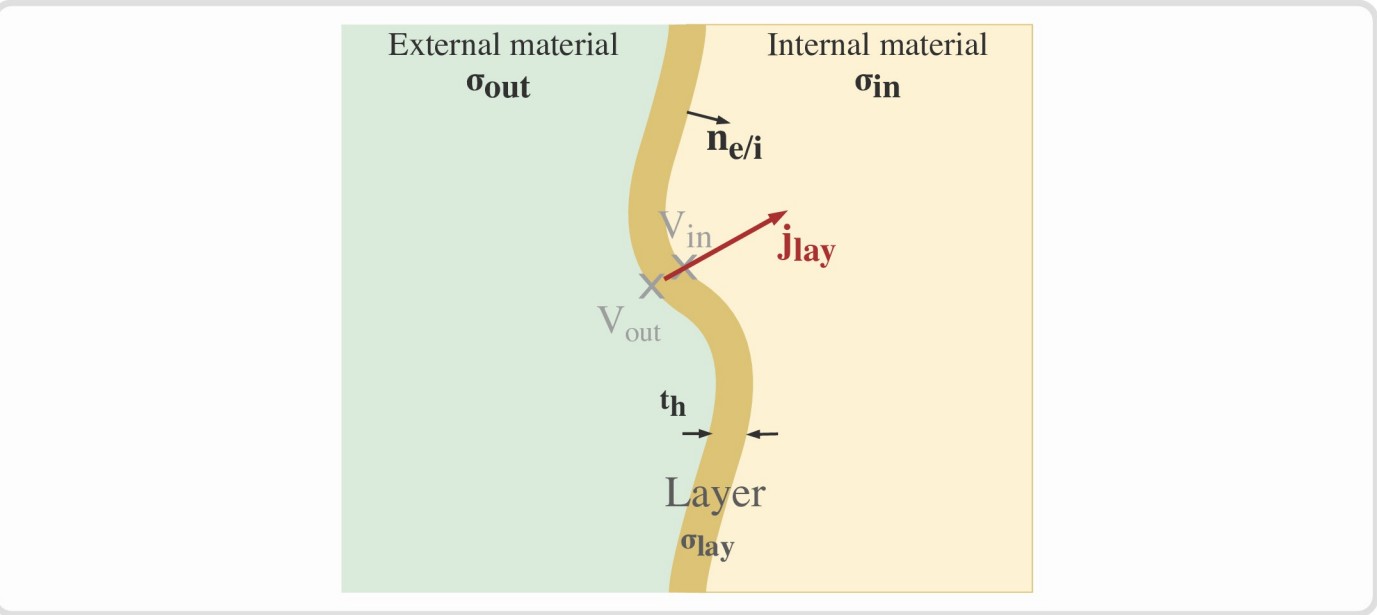

**Fig 3. Thin-layer approximation in NRV.** Thin-layer of thickness $t_h$ and conductivity $\sigma_{lay}$, bounding two materials, internal and external, of conductivity $\sigma_{in}$ and $\sigma_{ext}$ respectively.

The eCAP recording is performed automatically for the user when instantiating a `recorder`-object (see Fig 2), which links one material with one or multiple `recording-points`-objects. `recording-points`-objects represent positions in space where the extracellular is recorded during the simulation. Using again space and time decoupling, the eCAP electrical potential at a position **r** at a time $t$ is computed as:

$$V_{eCAP}(\mathbf{r}, t) = \sum_{k \in \mathcal{A}} \sum_{i \in \mathcal{N}} I_{\text{mem } k,i}(t) V_{\text{footprint } k,i} \tag{13}$$

where $\mathcal{A}$ is the set of axons in the simulation, $\mathcal{N}$ is the set of computational nodes in the axon implementation (see nmod section below), $I_{\text{mem } k, i}$ the membrane current computed in the nmod section(see below) and $V_{\text{footprint } k, i}$ is a scalar. From a numerical perspective, Eq 13 is equivalent to a sum of dot products between two vectors: the membrane current computed in the nmod section of NRV (see below) and a recorder footprint. The footprint is computed only once for each axon in the nerve geometry before any simulation.

The footprint for one position $\mathbf{r_{k,i}} = (x_{k,i}, y_{k,i}, z_{k,i}) \in \mathbb{R}^3$ in space corresponding to the node $i$ of the axon $k$ for a `recording-points`-object at the position $r_{rec} = (x_{rec}, y_{rec}, z_{rec}) \in \mathbb{R}^3$ is computed either with PSA:

$$\begin{cases} V_{\text{footprint } k,i} = \dfrac{1}{4\pi \sqrt{\sigma_{yy}\sigma_{zz}x_d^2 + \sigma_{xx}\sigma_{zz}y_d^2 + \sigma_{xx}\sigma_{yy}z_d^2}} \\[2mm] x_d = (x_{k,i} - x_{rec}) \\[2mm] y_d = (y_{k,i} - y_{rec}) \\[2mm] z_d = (z_{k,i} - z_{rec}) \end{cases} \tag{14}$$

for anisotropic or isotropic materials ($\sigma = \sigma_{xx} = \sigma_{yy} = \sigma_{zz}$), of with LSA for isotropic materials only [76]

$$\begin{cases} V_{\text{footprint } k,i} = \dfrac{1}{4\pi\sigma\Delta l} \log \dfrac{\sqrt{h_i^2 + r_i^2} - h_i}{\sqrt{l_i^2 + r_i^2} - l_i} \\[2mm] \Delta l = |x_{k,i+1} + x_{k,l}| \\[2mm] r_i = \sqrt{(y_{k,i} - y_{rec})^2 + (z_{k,i} - z_{rec})^2} \\[2mm] h_i = |x_{k,i} - x_{rec}| \\[2mm] l_i = h_i + \Delta l \end{cases} \tag{15}$$

In both cases, the eCAP simulation is performed after the computation of neural activity, which is explained in the next section.

## nmod section: Generating and simulating axons

The description of a neuronal context in NRV and the computation of the trans-membrane potential are described in a hierarchical manner in Fig 4. At the bottom of the hierarchy, axons are individual computational problems for which the NRV framework computes the membrane voltage response to extracellular electrical stimulation. The conventional hypothesis is that each axon is independent of others, i.e. there is no ephaptic coupling between fibers, so all

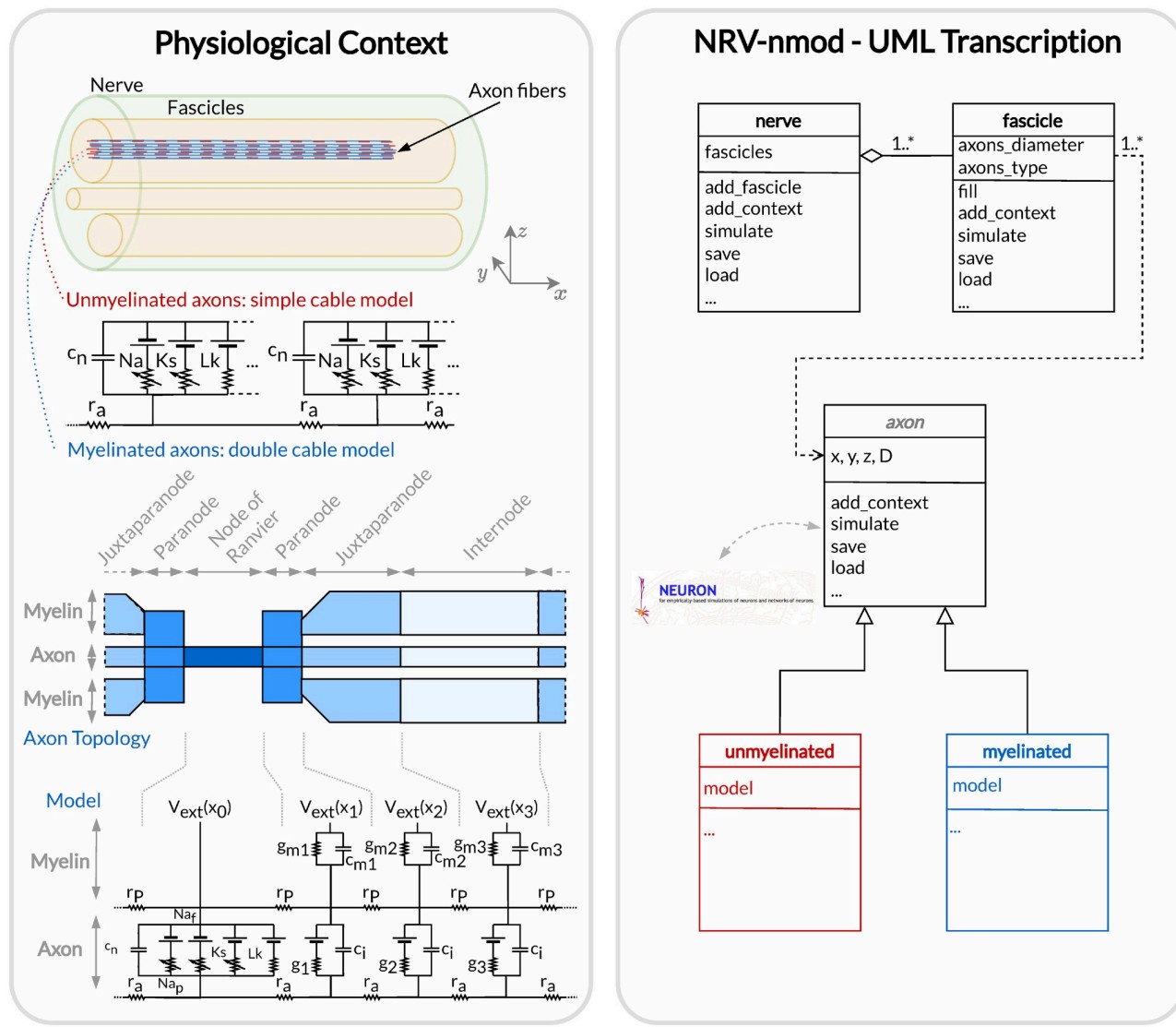

**Fig 4. Node section of the NRV framework.** Left: Overall view of the neuronal context of a generic PNS simulation. A nerve is an entity composed of one or multiple fascicles that do not overlap, each containing myelinated and/or unmyelinated axons. Right: NRV's transcription of the neuronal context with dedicated classes that can be combined and used by the framework to evaluate the resulting neural activity of each axon.

axon computations can be performed separately. From a computational point of view, this hypothesis transforms neural computation into an embarrassingly parallel problem, allowing massively parallel computations. In this section, details of the models are given using a bottom-up approach: axon models are described first, followed by fascicle entities, and finally nerves.

**Axon models.** Axonal fibers in NRV are defined with the `axon`-class. This class is an abstract Python class and cannot be called directly by the user. It however handles all generic definitions and the simulation mechanism. Axons are defined along the $x - axis$ of the nerve model. Axon (y,z) coordinates and length are specified at the creation of an `axon`-object. End-user accessible `Myelinated`-class and `unmyelinated`-class define myelinated and unmyelinated fiber objects respectively and inherit from the abstract `axon`-class.

Computational models can be specified for both the myelinated and unmyelinated fibers. Currently, NRV supports the MRG [43] and Gaines [45] models for myelinated fibers. It also supports the original Hodgkin-Huxley model [77], the Rattay-Aberham model [50], the Sundt model [48], the Tigerholm model [44], the Schild model [78] and its updated version [79] for unmyelinated fibers.

MRG and Gaines model's electrical properties are available on ModelDB [80] under accession numbers 3810 and 243841 respectively. Interpolation functions used in [45] to estimate the relationship between fiber diameter and node-of-Ranvier, paranode, juxtaparanodes, internode length, and axon diameter generate negative values when used with small fiber diameter. In NRV, morphological values from [43] and from [81] are interpolated with polynomial functions. Plots of the axon morphological properties and interpolation are provided in S2 Text and S2 File. Parameters of the unmyelinated models are taken from [82] and are available on ModelDB under accession number 266498.

The extracellular stimulations handled by the `fmod`-section of NRV are connected to the axon-object with the `attach_extracellular_stimulation`-method, linking the `extracellular_context`-object to the axon. Voltage and current patch-clamps can also be inserted into the axon model with the `insert_V_Clamp`-method and `insert_I_Clamp`-method. The `simulate`-method of the axon-class solves the axon model using the NEURON framework. NRV uses the NEURON-to-Python bridge [47] and is fully transparent to the user. The `simulate`-method returns a dictionary containing the fiber information and the simulation results.

**Fascicle construction.** The `fascicle`-class of NRV defines a population of fibers. The `fascicle`-object specifies the number of axons in the population, the fiber type (unmyelinated or myelinated), the diameter, the computational model used, and the spatial location of each axonal fiber that populates the fascicle.

An axon population can be pre-defined and loaded into the `fascicle`-object. Third-party software such as AxonSeg [83] or AxonDeepSeg [84] can be used for generating axon populations from a histology section and loaded into the `fascicle`-object. Alternatively, the NRV framework provides tools to generate a realistic *ex novo* population of axons. For example, the `create_axon_population`-function creates a population with a specified number of axons, a proportion of myelinated/unmyelinated fibers, and statistics for unmyelinated and myelinated fibers' diameter repartition. Statistics taken from [85–87] have been interpolated and predefined as population-generating functions. Original distribution and interpolation functions are presented in S3 Text and S3 File. An axon-packing algorithm is also included to place fibers within the fascicle boundaries. The packing algorithm is inspired by [88] and the concept is illustrated in Fig 5.

The `fascicle`-class can perform logical operations (remove geometrically overlapping populations, diameter filtering) and geometrical operations (translations and rotations) on the axon population. Node-of-Ranvier of the myelinated fiber can also be aligned or randomly positioned in the fascicle. An `extracellular_context`-object is added to the `fascicle`-object using the `attach_extracellular_stimulation`-method. Intracellular stimulations can also be attached to the entire axon population or to a specified subset of fibers. The `simulate`-method creates an axon-object for each fiber of the fascicle, propagates the intracellular and extracellular stimulations and recorders, and simulates each of them. Parallelization of `axon`-object simulation is automatically handled by the framework and fully transparent to the user. The simulation output of each axon is saved inside a Python dictionary or in a folder.

**Simulation top level: The nerve-object.** The top-level `nerve` class is implemented to aggregate one or more fascicles and facilitate association with extracellular context.

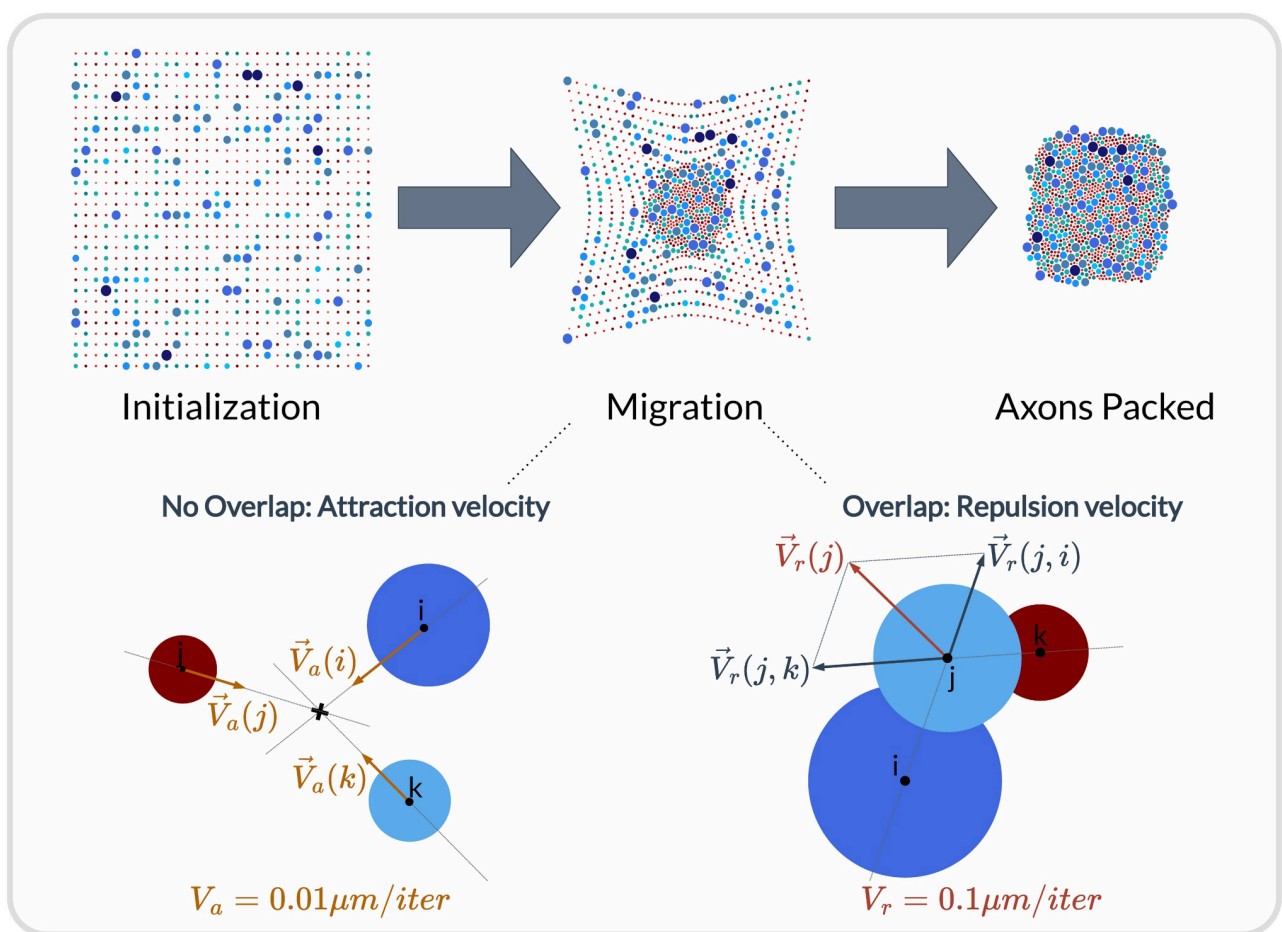

**Fig 5. Overview of the axon-packing algorithm inspired from [88].** The packing procedure was demonstrated with 1000 fibers. Myelinated axons are in blue and unmyelinated ones are in red. Fibers are randomly placed on a grid at initialization and migrate progressively toward the center. Each axon's new position is calculated at each iteration according to an attraction velocity or a repulsion velocity if axons are overlapping.

Fascicle-objects are attached to the nerve-object with the add_fascicle-method. The extracellular context is attached to the nerve-object and propagated to all fascicle-objects with the attach_extracellular_stimulation-method. The geometric parameters of the nerve-object and each fascicle-object are used to automatically generate the 3D model of the nerve (see Fig 2). Calling the simulate-method of the nerve-object simulates each fascicle attached to the nerve and returns either a Python dictionary containing all the results or only the simulation parameters, with the results saved in a specified folder.

## Optimize

Fig 6 describes the generic formalism adopted in NRV for running optimization algorithms on PNS stimulations. The optimization problem, defined in a Problem-class, couples a Cost_Function-object, which evaluates the cost of the problem based on user-specified outcomes (e.g., stimulus energy, percentage of axon recruitment, etc.), to an optimization method or algorithm embedded in the Optimizer-object. The optimization space is defined by specifying in the problem definition the subset of available adjustable simulation parameters (e.g.,

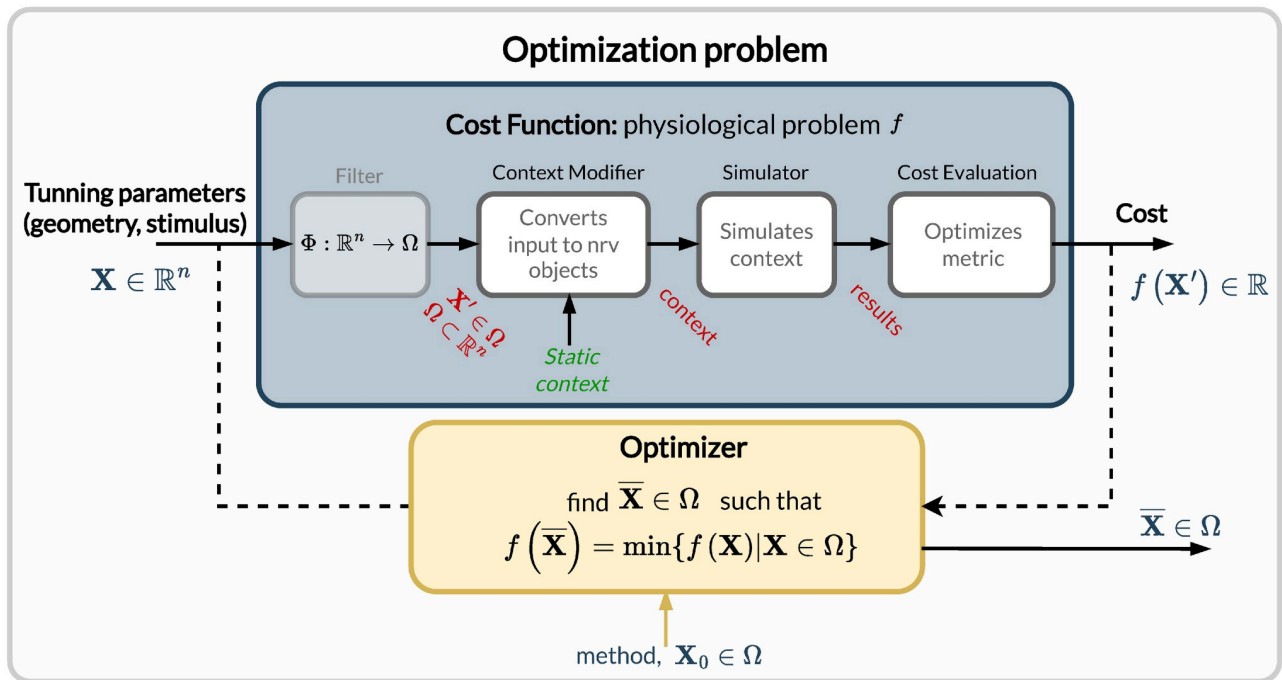

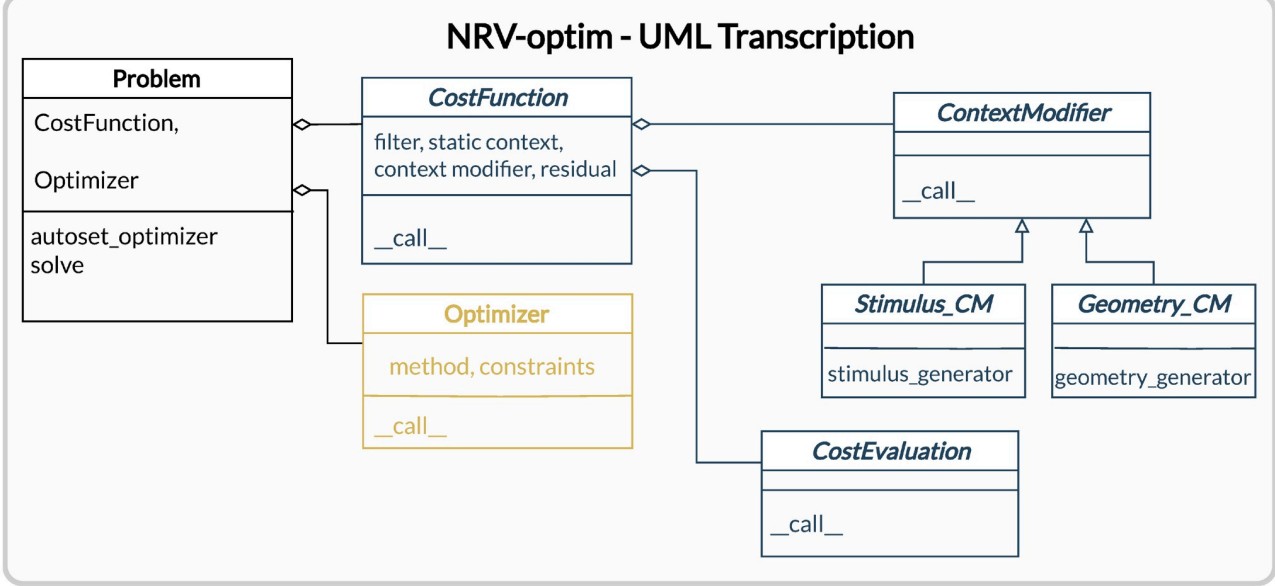

**Fig 6. Overview of optimization problem formalism and implementation in NRV.** A problem is described by combining a cost function and an optimizer or optimization method. The cost function transforms a vector of inputs to a cost by modifying a defined static simulation (of an axon, fascicle, or nerve) and by computing a user-defined cost from the simulation result. Corresponding classes have been developed to ease the formulation of problems.

stimulus shape, electrode size, etc.) and, optionally, their respective bounds. NRV provides methods and objects to construct the `Cost_Function`-object according to the desired cost evaluation method and optimization space. Specifically, the `Cost_Function`-class is constructed around four main objects (Fig 6):

- A filter: which is an optional Python `callable`-object, for vector formatting or space restriction of the optimization space. In most cases, this function is set to identity and will be taken as such if not defined by the user.

- A static context: it defines the starting point of the simulation model to be optimized. It can be any of the `nmod`-objects (axon, fascicle, or nerve) to which all objects describing stimulation, recording and more generally the physical context are attached.

- A `ContextModifier`-object: it updates the static context according to the output of the optimization algorithm and the optimization space. The `ContextModifier`-object is an abstract class, and two daughter classes for specific optimization problems are currently predefined: for stimulus waveform optimization or for geometry (mainly electrodes) optimization. However, there is no restriction to define any specific optimization scenario by inheriting from the parent `ContextModifier`-class.

- A `CostEvaluation`-object: uses the simulation results to evaluate a user-defined cost. Some examples of cost evaluation are included in the current version of the framework. Nonetheless, the `CostEvaluation`-class is a generic Python `callable`-class, so it can also be user-defined.

Optimization methods and algorithms implemented in NRV rely on third-party optimization libraries: SciPy optimize [89] for continuous problems, Pyswarms [90] as Particle Swarms Optimization metaheuristic for high-dimensional or discontinuous problems.

## Results

### FEM models cross-validation

While COMSOL Multiphysics is frequently used in hybrid modeling [33, 34, 40, 49], only the TxBDC Cuff Lab solution is based on the FEniCS solver [52]. The implementation of the FEM equations and the thin-layer approximation are first validated with a passive bi-domain 2-D model. Details and results of the validation are provided in the supporting information (S4 Text). In the rest of this section, we evaluate the effect of the selected FEM solver (FEniCs or COMSOL Multiphysics) on the calculation of the electrode footprint and the resulting Rattay activation function [6]. We also evaluate the effect on the activation thresholds of the axon, as the activation function is a simple but imprecise predictive approach [91, 92]. Python scripts for generating and plotting the data presented in this section are available in the supporting information (S1 Archive).

**Electrode footprint computation.** The extracellular voltage footprint generated by a LIFE and by a monopolar circular cuff-like electrode along an axonal fiber located at the center of a monofascicular nerve are compared between results obtained with COMSOL Multiphysics and with FEniCs. LIFE active-site is $1000\mu m$ long and $25\mu m$ in diameter. Cuff-like electrode contact width and thickness are $500\mu m$ and $100\mu m$ respectively. The LIFE active site is placed $100\mu m$ away from the fiber inside the fascicle. The cuff electrode fully wraps the nerve and has a thin insulating layer on its outer surface to restrict current flow. The extracellular voltage footprint along the fiber, as well as its spatial second derivative [6], are shown in Fig 7a and 7b for the LIFE and the cuff-like electrode respectively. The relative difference between the two solvers' estimation is also plotted.

COMSOL Multiphysics and FEniCS estimated footprints are in very good agreement for both electrodes. The maximum electrode footprint difference is about 1.4% with LIFE and about 1.7% with a cuff-like electrode. The difference peaks at the maximum of the footprints,

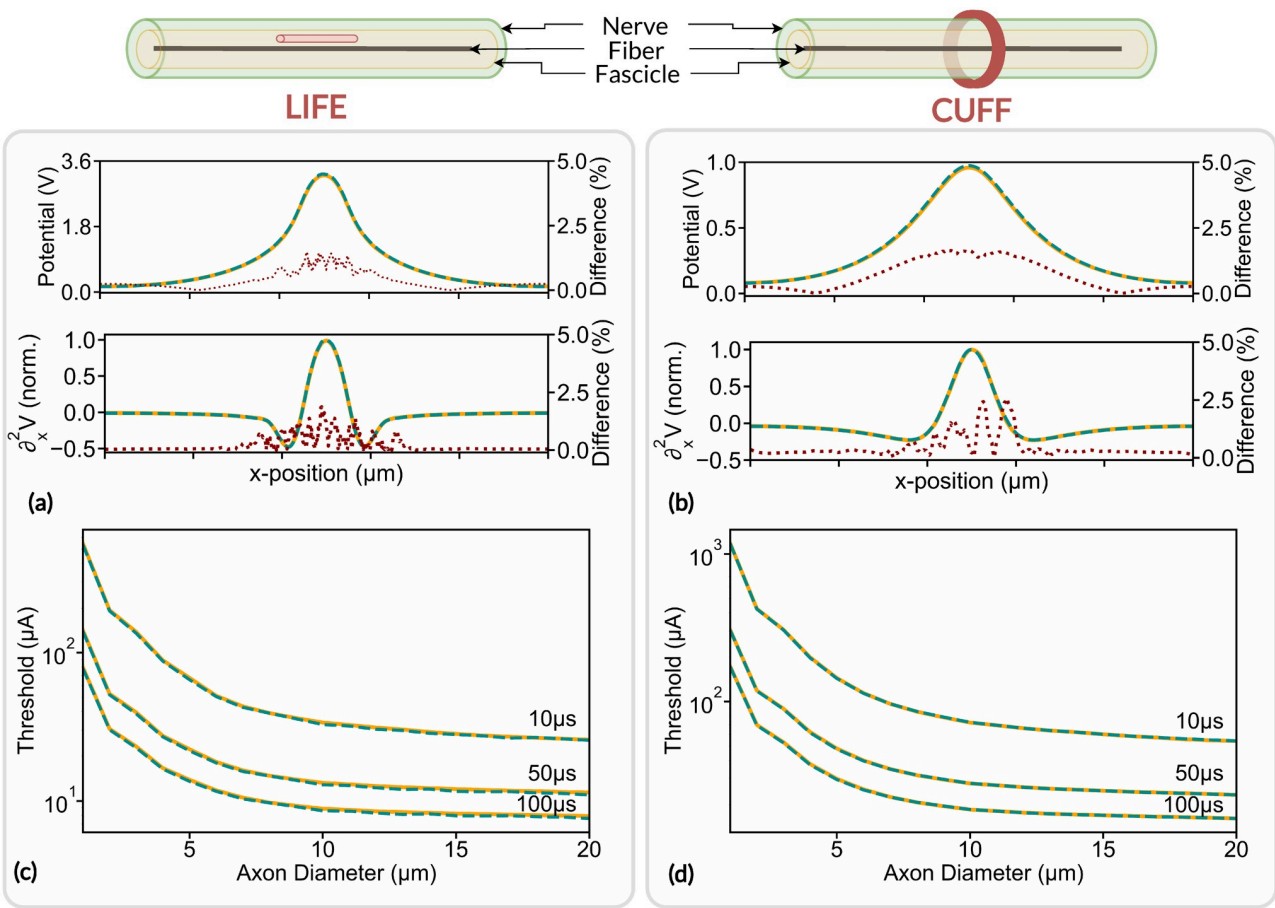

**Fig 7. Electrode footprints and axon activation thresholds estimation.** Footprints and activation thresholds are evaluated using the FEniCS solver (orange line) and COMSOL Multiphysics (dashed blue line) with a LIFE ((a) and (c)) and a cuff-like electrode ((b) and (d)). The relative difference in electrode footprint estimation is plotted in a red dotted line. The nerve and the fascicle are $1000\mu m$ and $800\mu m$ in diameter respectively. The fascicle is surrounded by a $25\mu m$ thick layer of perineurium, which corresponds to about 3% of the fascicle diameter [18]. The nerve is plunged in a $10mm$ large saline bath (not represented). The fascicle is made of anisotropic bulk endoneurium ($\sigma_x = 0.57S/m$, $\sigma_y = 0.085S/m$) and epineurium ($\sigma = 0.085S/m$). Perineurium and saline conductivity are set to $\sigma = 0.002/m$ and $\sigma = 2S/m$ respectively [66, 67].

i.e. where the axonal fiber is the closest to the electrodes. Dissimilarities near the electrodes' active site are to be expected as both FEM solvers are likely to handle boundary conditions slightly differently. The induced neural activity in axonal fiber is related to the spatial second derivative of the electrode footprint [50], which differs by a maximum of 1.9% with LIFE and a maximum of 2.4% with cuff electrode between COMSOL multiphysics and FEniCS estimations.

**Activation thresholds.** This section evaluates the impact of the difference in electrode footprint estimation between FEniCS and COMSOL Multiphysics on the induced neural dynamics. Specifically, the influence on the activation threshold, i.e. the minimum stimulation current required to induce an action potential within an axonal fiber, is evaluated. The axonal fiber is modeled as a myelinated fiber using the MRG model [43] located at the center of the monofascicular nerve. The activation threshold is evaluated for an axon diameter ranging from $1\mu m$ to $20\mu m$. The axon is stimulated with a monophasic pulse with a pulse duration of $10\mu s$, $50\mu s$, and $100\mu s$ delivered to the nerve with a LIFE and a cuff-like electrode. LIFE and

cuff-like electrode footprints were computed with both COMSOL Multiphysics and FEniCS and the activation thresholds were estimated with the binary search algorithm implanted in the framework, with a search tolerance of 0.1%. Results are illustrated in Fig 7c and 7d for the LIFE and cuff-like electrode respectively.

Thresholds estimated with FEniCS match very well those estimated with COMSOL Multiphysics, for both the LIFE and the cuff electrode. The maximum threshold estimation difference is about 1.9% with the LIFE and about 0.5% with the cuff electrode. Mean differences are about 0.08% and 0.3% respectively which confirms and validates the implementation of FEniCS in NRV.

### Replication of previous results

**Replication of *in silico* studies.**   The NRV framework aims at facilitating the reproduction of *in silico* studies. As an example, we replicated in this section a study on the high-frequency alternating current (HFAC) block mechanism modeled by Bhadra *et al.* [31]. The study demonstrated the phenomenon of HFAC nerve conduction block in a mammalian myelinated axon model and characterized the effect of HFAC frequency, electrode-to-axon distance, and fiber diameter on the HFAC block threshold. The axon fiber dynamics were also studied to better understand the underlying mechanism.

Here the study is duplicated as closely as possible using the amount of information available in the published article. A 51 Node-of-Ranvier fiber was modeled using the MRG model [43]. A PSA electrode was placed over the central node of the fiber and placed in a homogeneous isotropic medium. The conductivity of the medium was set to $0.2 S/m$. The neural conduction was verified with a test action potential generated with an intracellular current clamp at the myelinated fiber's first node of Ranvier. The block threshold was defined as the minimum HFAC peak-to-peak amplitude required to block the propagation of the test action potential and was estimated with a binary search method with a search tolerance of 1%.

The effect of the electrode-to-axon distance and the HFAC frequency is plotted in Fig 8a and 8b respectively. The neural dynamic of the axon's 25$^{th}$ node of Ranvier is shown in Fig 8c. Measured block thresholds are very close to the one obtained in the original study from Bhadra *et al*. The dynamics of the gating particles and the membrane potential are also very similar: the membrane voltage is highly contaminated by high frequency near the electrode but less on both sides and the action potentials from the onset response are visible. As noted by Bhadra *et al*, particle values show a rapid evolution and a high value for *m*, small variations for *h* around a small average value, and slow evolution for *s* and *mp* to values near 1. The evolution of block threshold with distance and frequency also follows the original study.

As expected, no exact match was obtained as some parameters of the simulation, such as spatial discretization, had to be guessed. Also, the study of Bhadra [31] uses the original discrete model parameters of the MRG model, whereas we use interpolated values in our framework. This however validates the implementation of the neural models in the NRV framework. Python scripts to generate and plot this *in silico* study are made available in the supporting information (S2 Archive). Furthermore, the iPython notebook file provided in supporting information (S4 File) also demonstrated how in just a couple of lines of code this simulation can be replicated.

**Replication of *in vivo* studies.**   In this section, the *in vivo* experiment from Nannini and Horch [93] and from Yoshida and Horch [94] were replicated with the NRV framework. Both studies aimed at characterizing the recruitment properties of LIFEs implanted in the tibial nerve of a cat. We focused our efforts on replicating the fiber recruitment versus stimulation current curve (Fig 5a from [93]) and the fiber recruitment versus pulse-width (Fig 2 from

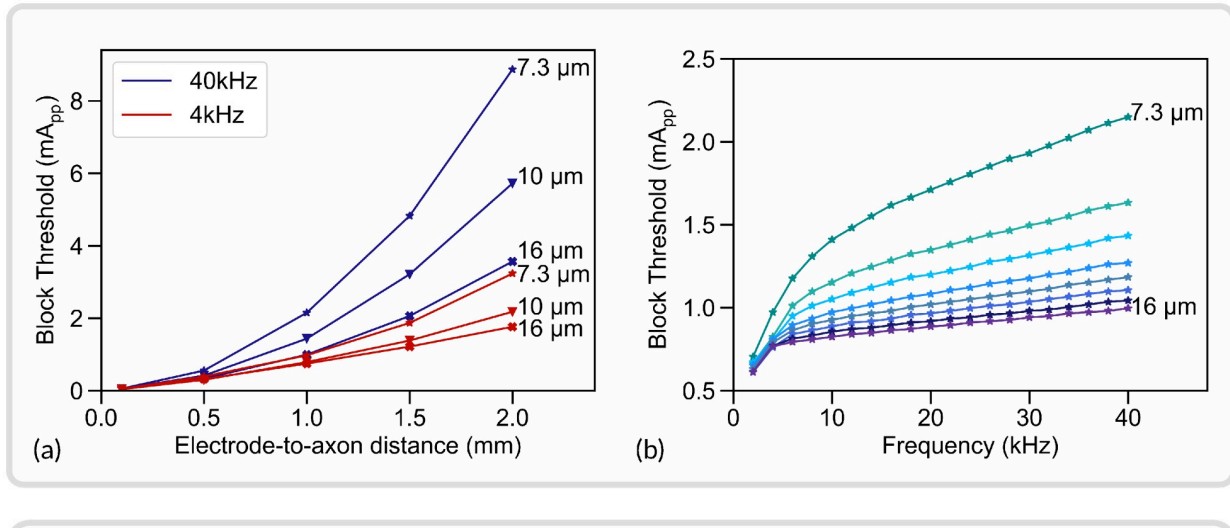

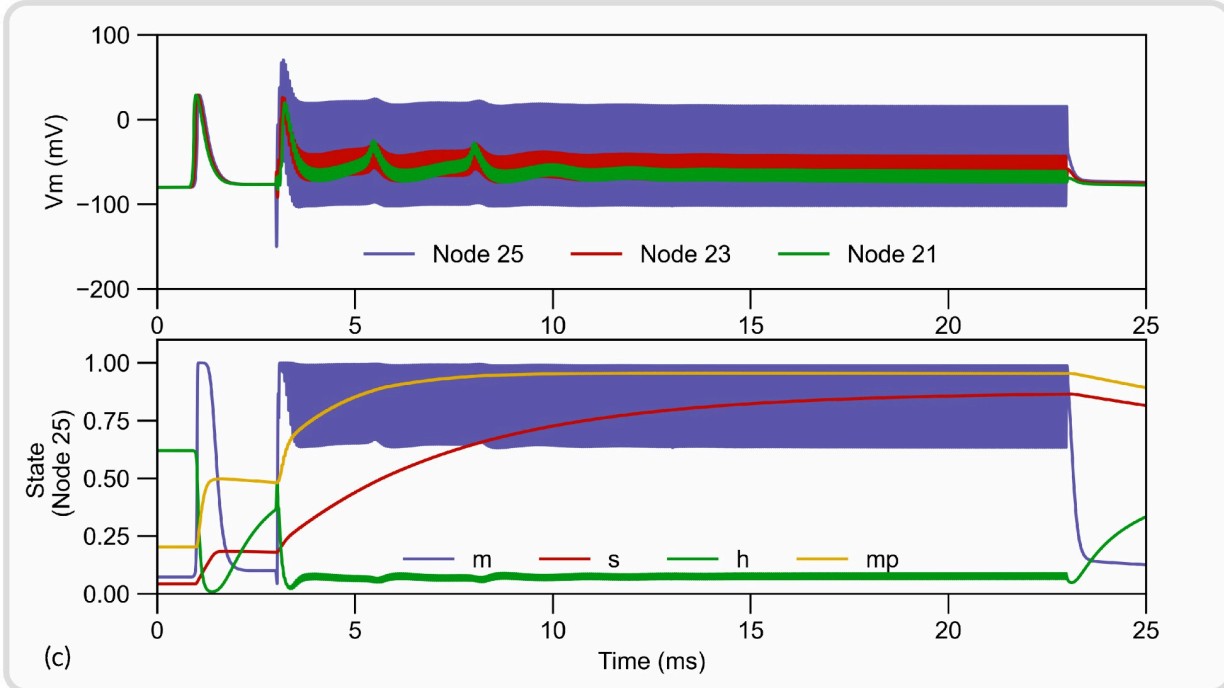

**Fig 8. Replication of an *in silico* study from [31] with NRV.** (a) Block thresholds versus electrode-to-axon distance for a $7.3\mu m$, a $10\mu m$ and a $16\mu m$ axon. Tested HFAC frequency is 4kHz and 40kHz; (b) Block thresholds versus HFAC frequency for axon diameters ranging from $7.3\mu m$ to $16\mu m$; (c) Membrane potential (top) and gating variables (bottom) of a $10\mu m$ axon. An action potential is initiated at $t = 0.5ms$ with an intracellular current clamp. A supra-blocking threshold HFAC is applied from $t = 3ms$ to $t = 23ms$.

[94]). In both studies, fiber recruitment was estimated from recordings of the force produced by the gastrocnemius muscle (GM). As a first approximation, the cat's tibial nerve was modeled as a mono-fascicular nerve. The fascicle innervating the GM muscle measures approximately one-fourth of the tibial nerve in diameter in the rat [95, 96] and the tibial nerve of the cat has a diameter of approximately 2.2mm [97]. Thus, the simulated GM innervating fascicle diameter was set to $550\mu m$. The fascicle was filled with 500 randomly picked myelinated axons

with diameters ranging from $1\mu m$ to $16\mu m$. Axon diameter probability density was defined from fiber diameter distribution measured in the cat's tibial nerve [98]. LIFEs were modeled following the given specification, i.e. with a diameter of $25\mu m$ and an active-site length of $1mm$. Neither study provided precise information on the electrode location within the fascicle. Notably, no post-experimental histological studies were carried out around the electrode implantation zone. Consequently, *in silico* recruitment curves were estimated with 10 different LIFEs locations randomly positioned (uniform distribution) within the fascicle. Simulated recruitment curves with original data points superimposed are shown in Fig 9. We also define the recruitment rate as the slope of the curve evaluated between 0.2 and 0.8 of the recruitment.

Simulated recruitment curves and *in vivo* experimental data are in good agreement: reducing the pulse width from $50\mu s$ to $20\mu s$ decreases the recruitment rate by a factor of about 2.1 in the Nannima and Horch study and by about 2.4 on average in the *in silico* study ($p < 0.01$, t-test) as shown in Fig 9a. The stimulation amplitude does not significantly affect the recruitment rate in the simulated recruitment versus pulse width duration curves ($p > 0.05$, t-test), and the recruitment rate is increased by a factor of 1.1 only when the stimulation amplitude is increased from $7\mu A$ to $9\mu A$ in the Yoshida and Horch study (Fig 9b). When the stimulation pulse width is 1ms, the *in vivo* recruitment difference between a $7\mu A$ and a $9\mu A$ pulse amplitude is about 0.37 and is about 0.48 in the *in silico* study ($p < 0.01$, t-test). Effects of pulse duration and pulse amplitude observed in the *in vivo* are well captured by the simulation, but the model overestimated the effect of pulse amplitude by about 22% and the effect of pulse width by about 14% when compared to the *in vivo* data. Both *in vivo* studies do not provide any statistical data thus the comparison is limited.

The absolute predictions of the *in silico* recruitment curves follow reasonably closely *in vivo* data. Dissimilarities are however observed, notably in the lower range of the stimulation current versus recruitment curves (Fig 9a). *In silico* studies also underestimate the recruitment rate by about 53% on average when compared to the Nannima and Horch study (Fig 9a) and by about 23% on average when compared to the Yoshida and Horch study (Fig 9b). Such differences are to be expected due to the different nature of *in vivo* and *in silico* recruitment

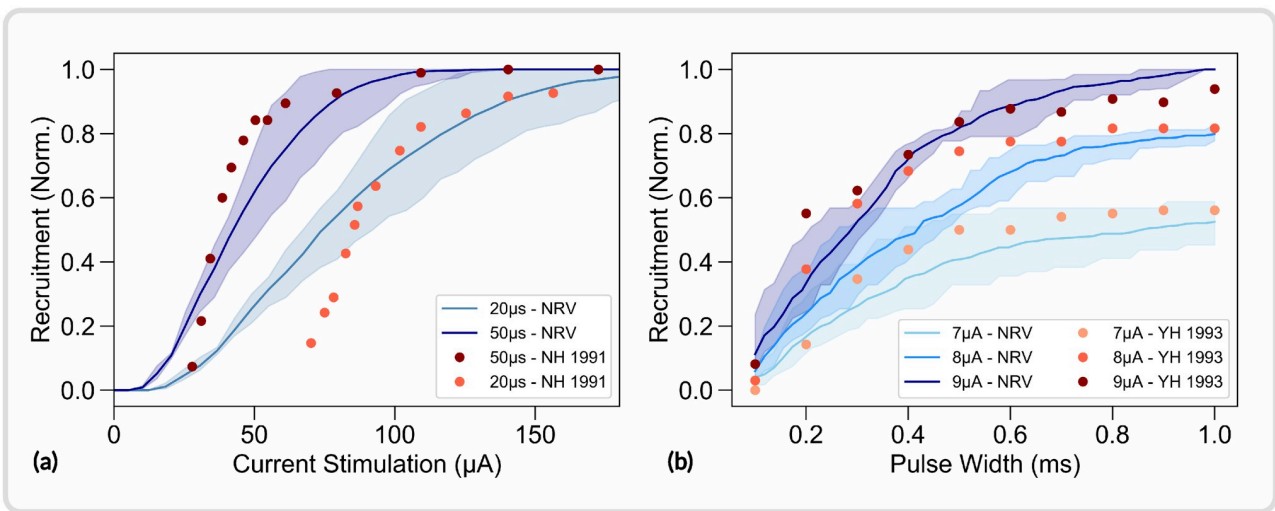

**Fig 9. Replication of *in vivo* studies from [93] and [94] with NRV.** *In silico* and *in vivo* data superimposed. *In silico* fibers are myelinated only and modelized using the MRG model [43]. (a) Recruitment versus stimulation current with a $50\mu s$ and $20\mu s$ cathodic pulse duration; (b) recruitment versus cathodic duration with a $7\mu A$, $8\mu A$, and $9\mu A$ stimulus; YH 1993 refers to data from [94]; NH 1991 refers to data from [93]. All experimental data are normalized with the maximum measured force for comparison purposes.

curves. *In vivo* recruitment data are obtained from muscle force recording that is innervated by a specific and localized subset of fiber in the tibial fascicle. *In silico* curves are obtained from the entire fiber population without spatial or functional considerations, resulting in much smoother recruitment curves.

This example also demonstrates the influence of LIFEs positioning on recruitment curves. The random positioning of LIFEs led to a standard deviation of the simulated recruitment rates of about on average 21%. This result indicates that the *in vivo* evaluation of other stimulation parameters effect could be hidden by the variability caused by the uncontrolled positioning of LIFEs during implantation. *In silico* studies using the NRV framework facilitate the investigation of isolated stimulation parameters' influence as each parameter can be precisely controlled and monitored. Python scripts for generating and plotting the data presented in this section are available in the supporting information (S3 Archive).

## Extracellular recordings

To demonstrate the extracellular recording capabilities of NRV, we simulated 5 monofascicular nerves comprised of 500 myelinated axons and 1059 unmyelinated axons randomly distributed according to the statistics for the sciatic nerve [98]. The corresponding radius of generated fascicles is about 120$\mu m$. 8 recording points are placed along the fascicle at the surface of the fascicle, each 2.5$mm$, starting at 2.5$mm$ from a current clamp initiating an action potential on all fibers at the same time, as illustrated in Fig 10.

An example of evoked compound action potential (eCAP) on the 8 successive recording points is shown in Fig 10a. Two propagating eCAPs are visible, highlighted in blue and red in the figure, and correspond to the fast propagation of action potentials in myelinated fibers and the slow propagation in unmyelinated fibers, respectively. The voltage potential resulting from the stimulation artifact is also visible. The graph shows a linear relationship between amplitude and latency (in log scale), which was also observed *in vivo* [99]. The *in silico* amplitude-latency slope for the myelinated fibers is about -2.0 indicating an inverse square relationship. The *in vivo* study reported a slope of about -1.8, which is consistent with the *in silico* estimation. It should be noted that points on the first *in silico* recording point are not included in the fit. For myelinated eCAP, the amplitude is too small and likely obscured by the stimulation artifact. For unmyelinated eCAP, the amplitude is large, due to full synchronization of the fibers, which is unlikely to be observed in *in vivo* experiments and inherent to *in silico* simulations. Python scripts for generating and plotting the data presented in this section are available in the supporting information (S4 Archive).

## Optimizing the stimulus energy

To illustrate the optimization capabilities implemented in the NRV framework, we set up an optimization problem aimed at reducing the stimulus energy of an electrical stimulation. The problem is illustrated in Fig 11a. The static context of the optimization problem consisted of a monofascicular nerve with a LIFE implanted in its center. We arbitrarily set the fascicle diameter to 200μm and filled it with myelinated axons until it was fully packed. This resulted in a population of 205 fibers after removing those that overlapped with the LIFE. Each fiber was modeled using the MRG model. The cost function of the optimization problem is defined as the sum of a stimulus energy contribution [59] and the number of recruited fibers contribution:

$$Cost = \alpha_e \sum_{t_k} i^2(t_k) + \alpha_r(N_{axon} - N_{recruited}) \tag{16}$$

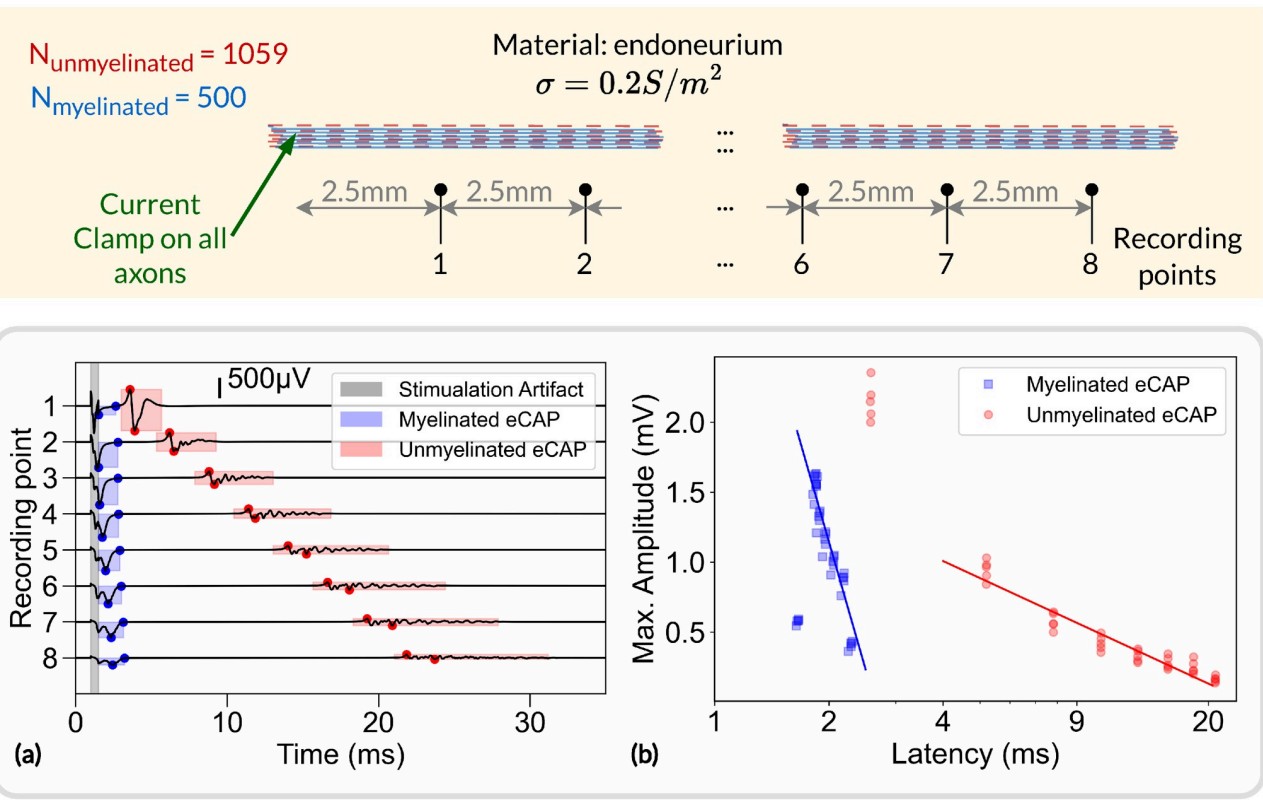

**Fig 10. Extracellular recordings simulated with NRV.** Top: schematic representation of the simulation. Myelinated and unmyelinated fibers are placed in a homogeneous infinite medium (endoneurium) and activated with a current clamp. Recording electrodes are placed along the fibers and separated by 2.5*mm*; (a) simulated potential at each recording point. Myelinated and unmyelinated eCAP contributions are highlighted in blue and red, respectively. The stimulation artifact is highlighted in gray; (b) Maximum amplitude of myelinated eCAP (in blue) and unmyelinated eCAP (in red) versus eCAP latency.

Where $t_k$ is the discrete time step of the simulation, and $\alpha_e$ and $\alpha_r$ are two weighting coefficients. We chose $\alpha_r >> \alpha_e$ to favor the recruitment of fiber over the stimulus energy reduction in the optimization process.

The stimulation pulse is adjusted via the NRV's context modifier according to two scenarios:

- The stimulus is a cathodic conventional square pulse. In this scenario, both the pulse duration and pulse amplitude can be optimized, resulting in a two-dimensional optimization problem. The tuning parameters input vector $\mathcal{X}_{sq}$ of the optimization problem is thus defined as follow:

$$\mathcal{X}_{sq} = \begin{pmatrix} I_{sq} \\ T_{sq} \end{pmatrix} \quad (17)$$

Where $I_{sq}$ and $T_{sq}$ are the conventional pulse amplitude and pulse width respectively.

- The stimulation is defined as an arbitrary cathodic pulse through interpolated splines over $N$ points which are individually defined in time and amplitude [101]. This second optimization

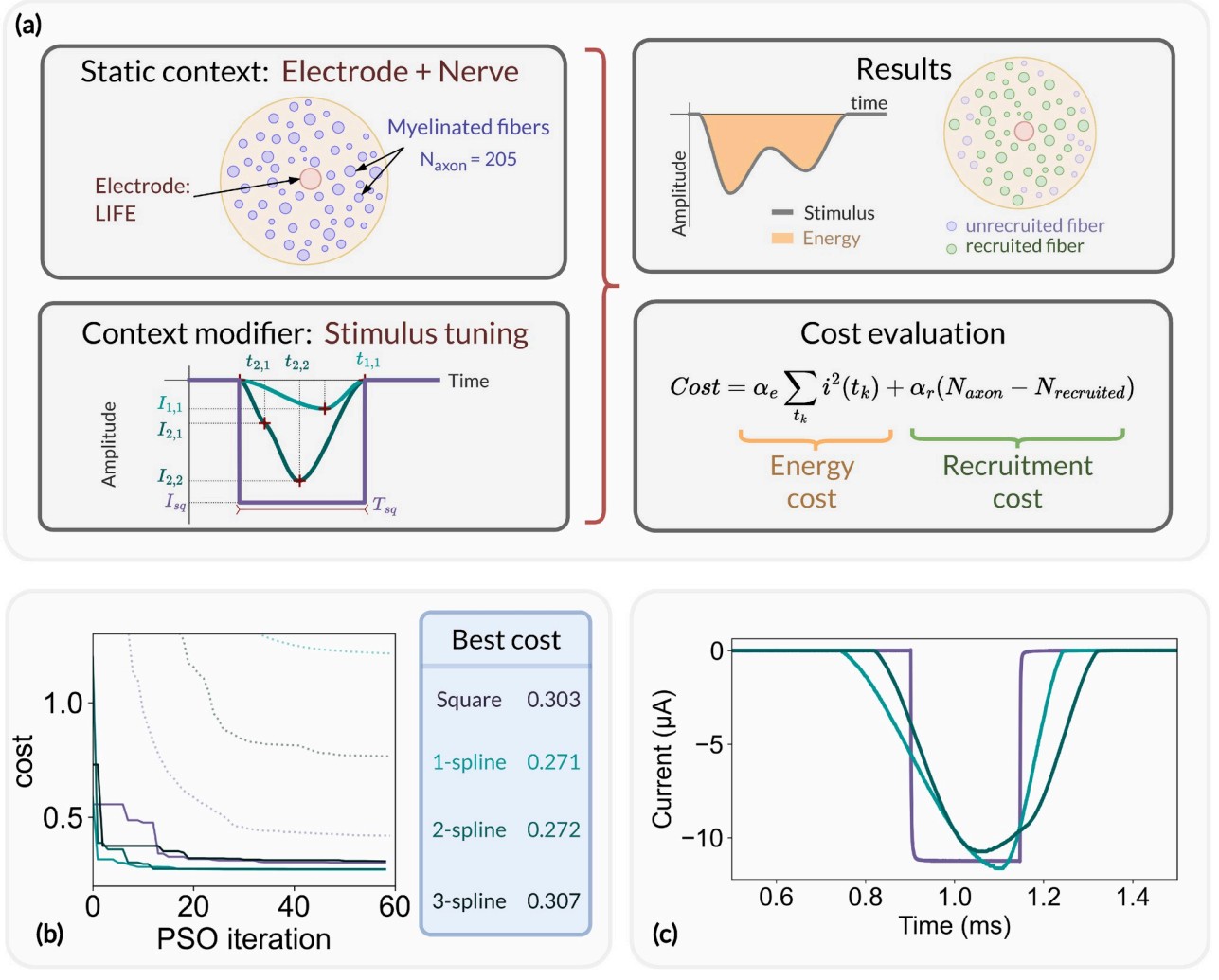

**Fig 11. Optimization problem formulation and results.** (a) Overview of the optimization problem. The simulation context consists of a monofascicular nerve and a LIFE and the stimulus parameters are modified by the context modifier after each optimization iteration (left). The simulation results are processed to estimate the number of recruited fibers as well as the stimulus energy and then used to update the cost function (right). (b) Cost function evolution: best (solid line) and average (dotted line) cost of the swarm for a square pulse (purple), a 1- (light blue), a 2- (teal blue), and a 3- (dark blue) spline stimulus; (c) Current recording of the square pulse, 1- and 2-splines energy-optimal stimulus applied to a LIFE plunged into a saline solution using a custom arbitrary-waveform stimulator [100]. The current is recorded via a $1k\Omega$ shunt resistor.

scenario results in a 2*N*-dimensional problem with the input vector $\mathcal{X}_{s_N}$ defined as follow:

$$\mathcal{X}_{s_N} = \begin{pmatrix} I_{s_1} \\ t_{s_1} \\ \vdots \\ I_{s_N} \\ t_{s_N} \end{pmatrix} \quad (18)$$

Where the pair of elements $(I_{s_n}, t_{s_n})$ represents the simulation amplitude and time coordinates of each interpolated point.

The optimization problem was solved for the square pulse stimulus and for a 1-, 2-, and 3-point spline interpolation stimulus, resulting in four optimization problems with 2, 2, 4, and 6 dimensions respectively. A particle swarm optimization (PSO, [102]) algorithm was used as the solver, with 25 PSO particles and 60 iterations.

For each optimization scenario, the evolution of the best cost and average cost of the PSO particles is shown in Fig 11b. After the first iteration of the PSO, the average cost for each tested stimulus is greater than 1, indicating that, on average, not every fiber of the fascicle is recruited. However, at least one particle of the PSO starts with stimulus parameters that result in a cost below 1 and thus fully recruits the fascicle. In such a situation, further reduction of the cost function can only be achieved by reducing the energy of the stimulus. For square pulse stimulation, the best cost was reduced from 0.72 to 0.30 over the 60 PSO iterations, representing a cost reduction of around 40%. The square pulse energy-optimal parameters were found to be $T_{sq} = 232\mu s$ and $I_{sq} = 11\mu A$. It is worth noting that the $T_{sq}$ obtained is very close to the $200\mu s$ energy-optimal pulse duration found by Wongsarnpigoon and Grill [59]. The stimulus energy was further reduced with the 1-spline stimulus, resulting in an energy reduction of about 10% compared to the energy-optimal parameters of the square pulse. No significant differences were observed between the 1-spline and 2-spline stimuli, and the 3-spline stimulus showed no improvement in energy compared to the conventional square pulse.

Each aspect of the optimization problem was described using the NRV objects, methods, and functions described in this paper. In particular, the optimization was run on multiple CPU cores using the built-in parallelization capabilities of the framework's `Problem` and `CostFunction` classes. The parallelization resulted in a runtime for each optimization scenario of about 20 hours on a single core to about 1 hour on 50 CPU cores.

Finally, we exported the energy-optimal square pulse, 1-spline, and 2-spline stimuli to an arbitrary waveform neurostimulator developed by our group [100]. The stimuli were applied to a custom-made LIFE with similar geometric properties to those used in the static context of the optimization problem, plunged into a saline solution. The measured current for each stimulus is plotted in Fig 11c. The latter result demonstrates how *in silico* results from the NRV framework can be easily translated to *in vivo* experiments. All Python scripts and source files required to create and run the optimization problem, analyze the results, and transverse the results to the neurostimulator are available in the supporting information (S5 Archive).

## Framework validation

The integration of the FEM solver into the framework was validated by comparing the results with those obtained using the well-established commercial FEM software COMSOL Multiphysics. The difference in fiber stimulation thresholds was less than 2%, validating the implantation of the FEniCS-based FEM solver. It also shows that simulation results obtained with COMSOL Multiphysics and FEniCS can be safely compared and that migration from COMSOL Multiphysics to FEniCS does not change the simulation results.

NRV aims to facilitate the replication of other *in silico* studies and was demonstrated by replicating a study of KHFAC neural conduction block by Bhadra et al. [31]. Bhadra's results were closely replicated with NRV using only a few lines of Python code (supporting information S4 File). NRV provides a tool for other researchers to build on previously published work.

Recruitment curves predicted by NRV were compared with *in vivo* experiments published by Nannini and Horch and Yoshida and Horch [93, 94]. The *in silico* and *in vivo* recruitment curves were in line, and the simulation captured well the effect of pulse width and pulse

**Table 1. Comparison with other published open-source approaches.**

| | PyPNS | ASCENT | ViNERS | TxBDC | This Work |
|---|---|---|---|---|---|
| FEM Solver | external | external [1] | internal | internal | internal |
| Nerve Model | *ex novo* | histology *ex novo* | histology *ex novo* | *ex novo* | *ex novo* |
| Axon Pop. | *ex novo* | histology *ex- ovo* | histology *ex novo* | *ex novo* | histology *ex novo* |
| Axon Models | HH<br>MRG | MRG<br>RA<br>Sundt<br>TGH | Sundt Gaines | Passive | MRG<br>Gaines<br>HH<br>RA<br>Sundt<br>TGH |
| Electrodes | cuff | cuff [2] | cuff<br>array | cuff<br>intraneural | PSA<br>cuff<br>LIFEs |
| Stimulus | arbitrary | arbitrary | arbitrary | mono. pulse biph. pulse | arbitrary |
| Extracellular Recorders | analytic FEM | FEM [3] | FEM | none | analytic |
| Optimization | No | No | No | No | Yes |
| Language | Python | Python | MATLAB [4] | GUI | Python |
| Commercial Soft. | COMSOL [5] | COMSOL [4] | MATLAB [4] | None | None |
| Multiproc. | N/S | N/S | N/S | N/S | Yes |

N/S not specified; HH Hodgkin-Huxley model; MRG McIntyre-Richardson-Grill model; RA Rattay-Aberham model; TGH Tigerholm model.

[1] ASCENT automates communication between the external FEM solver (COMSOL Multiphysics) and the pipeline.

[2] ASCENT provides highly realistic cuff electrode models based on commercially available products.

[3] SCENT provides filter-based extracellular recording since version 1.3.0 [51].

[4] MATLAB, The MathWorks Inc, Massachusetts, USA.

[5] COMSOL Multiphysics, Comsol AB, Stockholm, Sweden.

amplitude on fiber recruitment profiles. The *in silico* study also demonstrated the effect of LIFE misplacement on recruitment curves, illustrating the advantages of *in silico* models for monitoring and assessing the influence of individual stimulation parameters. Results related to extracellular recording of neural fibers also successfully reproduced *in vivo* observed phenomena. *In silico* replication of *in vivo* is also a useful approach to extend data analysis and interpretation.

Our architecture, based on a translation of physical objects or contexts (neural fibers, fascicles and nerves geometries, and material properties. . .) to Python objects, also allows a generic description of a simulation context. This implementation, transparent to the user, has been successfully combined with optimization methods that allow automatic exploration of the parameter space and propose novel stimulation strategies.

## Comparison with other solutions

Table 1 provides a comparison with other open-source published solutions. From a modeling point of view, NRV makes the same simulation assumptions as most other approaches, i.e. there is no ephaptic coupling between axons, electro-diffusion effects are neglected, and the electric field in the nerve is solved under the quasi-static hypothesis.

In terms of functionality, NRV compares well with other published solutions. Commonly used myelinated and unmyelinated axon models are already implemented in the framework, and adding other published or custom axon models to NRV is straightforward. NRV handles multi-electrode stimulation with arbitrary waveform stimulation. Point-source electrodes are available for quick, geometry-free stimulation. Parameterized monopolar and multipolar cuff

electrodes and LIFEs are implemented in the framework for more realistic simulations. Like PyPNS, ViNERS, and ASCENT, NRV can simulate eCAP recordings. However, the recordings are calculated analytically, whereas PyPNS, ViNERS, and ASCENT provide more accurate FEM-based eCAP simulations. NRV is also the only published solution that integrates simulation optimization capabilities. Tuning stimulation parameters using optimization algorithms is a promising approach for improving PNS stimulation [34, 58–60, 103] and we believe that its integration into the framework is a great benefit for the end user. Also, we provide a solution where there is no restriction to which parameters can be optimized, and optimization methods can be added by contributors. Moreover, NRV was developed in close relation to arbitrary-waveform stimulation device [100]. In comparison with other approaches, we hope to provide complete solutions to explore stimulation strategies from *in silico* to *in vitro* or *in vivo* experimental setup and ease the development of future electroceuticals.

## Scalability: From basic usage to cluster computing

Like ViNERS and TxDBC, NRV is self-contained, i.e. the end user does not need to use any external software during the simulation. In addition, NRV does not depend on commercially licensed software such as MATLAB or COMSOL Multiphysics. NRV is based on Python, which is not only open-source but has grown in popularity over the last decade and is supported by a large community. Many Python packages are freely available, making the language more versatile than MATLAB. Python is also easier to deploy on any computer, cluster of computers, or supercomputer. The entire NRV framework, including third-party libraries and software, can be installed directly from the Pypi or Conda library managers. All third-party libraries used in NRV are open-source and versioned, greatly limiting potential future incompatibility or end-of-life issues. In contrast to solutions based on commercially licensed software, version retrograding of third-party libraries will remain possible in NRV, facilitating data reproducibility [56]. Another advantage of using the Python language is its ability to massively parallelize simulations without any license restrictions. NRV automatically handles the parallelization of simulations, which greatly reduces simulation times while being transparent to the end-user. Table 2 shows the execution times of NRV's key steps when simulating a monofascicular nerve filled with 100 myelinated axons on 4 different computing platforms.

**Table 2. Execution time for key NRV simulation steps.**

|  | Laptop 1 [1] |  | Laptop 2 [2] |  | Cluster 1 [3] |  | Cluster 2 [4] |  |
|---|---|---|---|---|---|---|---|---|
| CPU Cores | 1 | 4 | 1 | 4 | 1 | 48 | 1 | 48 |
| FEM Meshing (s) [5] | 1.2 | 0.5 | 0.6 | 0.2 | 1.6 | 0.5 | 0.8 | 0.3 |
| FEM Presolver (s) | 0.2 | 0.3 | 0.2 | 0.2 | 0.4 | 0.3 | 0.4 | 0.5 |
| FEM Solver (s) | 4.5 | 4.6 | 2.5 | 2.4 | 8.9 | 9.1 | 3.5 | 3.5 |
| FEM Dispatch (s) | 20.2 | 22.0 | 14.6 | 14.3 | 27.1 | 26.9 | 14.6 | 14.4 |
| Neuron Solver (s) | 193.1 | 71.0 | 104.5 | 26.6 | 316.1 | 17.1 | 204.5 | 15.6 |
| Total (s) | 219.1 | 98.4 | 122.5 | 44.2 | 354.6 | 59.2 | 223.2 | 35.3 |

timings are obtained from the simulation during 5ms of a monofascicular nerve with 100 myelinated fibers (MRG model [43]);

[1]Laptop running Sonoma 14.4 equipped with an Intel i5 CPU (4 cores) running at 1.4GHz with 16Go of RAM;

[2]Laptop running Sonoma 14.4 equipped with an Apple M1 Pro CPU (8 performance cores) running at 3.2GHz with 16Go of RAM;

[3]Cluster based on four Intel Xeon Gold 6230R CPUs (24 cores per CPU) at 2.10Ghz with 1To of RAM running Ubuntu 22.04;

[4]Cluster based on four AMD EPYC 7643 CPUs (48 cores per CPU) at 2.10Ghz with 1To of RAM running Ubuntu 22.04;

[5]8 CPU cores were allocated for meshing at most. Computing performance decreased when more CPU cores were used.

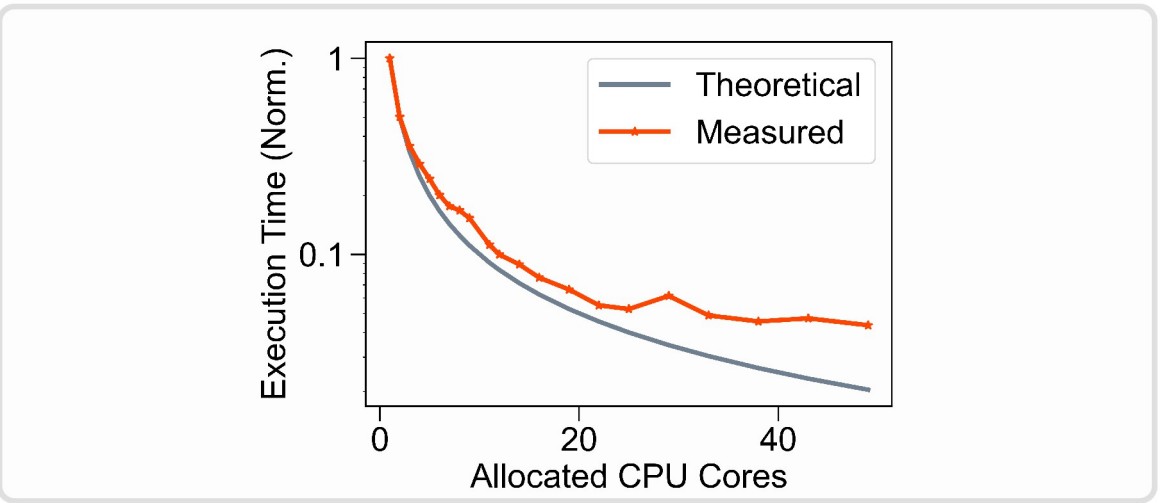

**Fig 12. Normalized NRV simulation execution time versus allocated CPU cores count.** 500 myelinated fibers (MRG model [43] are simulated during 5ms. Single CPU core execution time is used to normalize the data. The theoretical execution time is evaluated by dividing the single-core execution time by the number of allocated cores. Simulation is run on four Xeon Gold 6230R CPUs (104 CPU cores available) at 2.10Ghz with 1To of RAM running Ubuntu 22.04.

Each platform benefits from parallelization, with an overall execution time reduction ranging from 2.2 to 6.4. In particular, fascicle simulations are embarrassingly parallel problems, so the theoretical simulation execution time is inversely proportional to the available number of CPU cores. To validate this, we simulated 500 myelinated fibers on a computer cluster with an increasing number of allocated CPU cores and monitored the execution time (Fig 12). The parallelization significantly reduces the simulation execution time and follows the theoretical decrease up to 22 allocated CPU cores. Beyond 22 CPU cores allocated to the framework, no significant improvement in execution time is observed. This is likely due to thermal throttling effects or memory bandwidth limitations. However, in the case of multiple parallelized fascicle or nerve simulations, as with meta-heuristic optimizations, further reductions in computational cost can be expected with larger clusters.

## Availability and future directions

### Sources, distribution and community

NRV is a fully open-source framework developed in Python that aims to improve the accessibility of *in silico* studies for PNS stimulation. The framework uses the third-party open-source software Gmsh and FEniCS to solve the volume conduction problem and NEURON to simulate the neural dynamics of the fibers. The NRV sources and examples are freely available on GitHub (https://github.com/nrv-framework/NRV). The version of the framework associated with this manuscript is *v*1.1.0 (commit n˚ #1024: fb856d1). The framework is distributed under the CeCILL Free Software License Agreement. We hope to provide the research community with a tool for reproducible and fully open science. We also provide long-term support:

- Community for online assistance and code-sharing: https://github.com/nrv-framework/NRV/discussions and http://nrv-framework.org/forum/,

- Complete documentation and examples with granular complexity: https://nrv.readthedocs.io/en/latest/,

- Availability on package manager: https://pypi.org/project/nrv-py/.

More generally, all information about the framework is accessible on https://nrv-framework.org. The NRV framework is maintained using Continuous Integration/Continuous Development workflow, with a focus on retro-compatibility to ensure the reproducibility of scientific results.

### Perpectives

The NRV is an ongoing project that is constantly improving. To date, several improvements are under consideration. One of the limitations of the NRV framework in terms of functionality is the lack of support for histology-based realistic nerve geometry. Currently, NRV only provides classes and methods to generate parameterized cylindrical nerve and fascicle geometries, while tools to facilitate the simulation of realistic nerve geometry based on histological sections are integrated in ASCENT and ViNERS. However, no technical limitations of NRV prevent the future development of customized realistic nerves and electrodes. Since NRV is based only on open-source solutions supported by a consistent community, many different solutions and approaches can be explored to add this feature. The main challenge is to develop a solution that is easy to use for inexperienced users while being versatile enough to meet the needs of the community.

From a technical point of view, we will continue to optimize the code and extend the multi-processing capabilities to more features of the framework. We will also explore the use of hardware acceleration, such as GPUs, to further increase computational performance. The support of interactive visualization tools, which help to validate the different steps of the simulation process as well as to facilitate data analysis, will also be considered. From a scientific point of view, we currently work on emulating complex recording methods such as Electrode Impedance Tomography [104]. Another avenue is the simulation of interconnected networks and their connection to peripheral nerves to provide *in silico* models to more complex stimulation therapies such as vagus nerve stimulation for instance. An updated version of the framework roadmap is maintained online on the framework website https://nrv-framework.org/.

Updates and improvements to NRV will be continuously made available to the public on the dedicated GitHub repository. We hope that the growing number of users will increase the amount of relevant feedback that will help us to further improve the framework.

### Supporting information

**S1 Text. Stimulus in NRV.** Examples of stimulus creation and combination in NRV.
(PDF)

**S2 Text. Myelinated axon structural parameters.** Polynomial fit of myelinated axon structural parameters and original data.
(PDF)

**S3 Text. Axon diameter distributions available in NRV.** Description, plot, and reference of the axon diameter distributions available in NRV.
(PDF)

**S4 Text. Electrical physics validation.** Validation of the FEM equations and solver implementation on a 2-D bi-domain model.
(PDF)

**S1 File. Stimulus in NRV.** Python script to generate complex stimulus in NRV.
(PY)

**S2 File. Myelinated axon structural parameters.** Python script to visualize myelinated axon structural parameters.
(PY)

**S3 File. Axon diameter distributions available in NRV.** Python script to visualize axon diameter distributions available in NRV.
(PY)

**S4 File. Bhadra *et al. in silico* study replication with a iPython Notebook.** The iPython Notebook provides a didactic replication of the *in silico* study.
(IPYNB)

**S1 Table. Materials in NRV.** Table of available materials in NRV with corresponding references.
(PDF)

**S1 Archive. Comparison between FEniCS and COMSOL solvers.** Python scripts and data files to generate and plot the comparison of the two FEM solvers. The compared data are the electrode footprint (LIFE and cuff) and the resulting activation thresholds.
(ZIP)

**S2 Archive. Bhadra *et al. in silico* study replication.** Python scripts and data files to generate and plot the *in silico* study replication.
(ZIP)

**S3 Archive. Nannini and Horch, and Yoshida and Horch *in silico* study replication data.** Python scripts and data files to generate and plot the *in silico* study replication.
(ZIP)

**S4 Archive. Extracellular recordings *in silico* study.** Python scripts and data files to generate and plot the *in silico* extracellular study.
(ZIP)

**S5 Archive. Energy-optimal *in silico* study.** Python scripts and data files required to create and run the optimization problem, analyze the results, and translate the results to the neurostimulator.
(ZIP)

## Author Contributions

**Conceptualization:** Louis Regnacq, Olivier Romain, Yannick Bornat, Florian Kolbl.

**Data curation:** Thomas Couppey, Louis Regnacq, Florian Kolbl.

**Formal analysis:** Thomas Couppey, Louis Regnacq, Roland Giraud, Florian Kolbl.

**Funding acquisition:** Olivier Romain, Yannick Bornat, Florian Kolbl.

**Investigation:** Thomas Couppey, Louis Regnacq, Roland Giraud, Florian Kolbl.

**Methodology:** Thomas Couppey, Louis Regnacq, Roland Giraud, Olivier Romain, Yannick Bornat, Florian Kolbl.

**Project administration:** Florian Kolbl.

**Resources:** Florian Kolbl.

**Software:** Thomas Couppey, Louis Regnacq, Roland Giraud, Florian Kolbl.

**Supervision:** Olivier Romain, Yannick Bornat, Florian Kolbl.

**Validation:** Thomas Couppey, Louis Regnacq, Roland Giraud.

**Visualization:** Thomas Couppey, Louis Regnacq.

**Writing – original draft:** Thomas Couppey, Louis Regnacq, Florian Kolbl.

**Writing – review & editing:** Thomas Couppey, Louis Regnacq, Roland Giraud, Olivier Romain, Yannick Bornat, Florian Kolbl.

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
