## [Decision Letter · Decision Letter 0]

31 Mar 2024

Dear Dr Kolbl,

Thank you very much for submitting your manuscript "NRV: An open framework for in silico evaluation of peripheral nerve electrical stimulation strategies" for consideration at PLOS Computational Biology.

As with all papers reviewed by the journal, your manuscript was reviewed by members of the editorial board and by several independent reviewers. In light of the reviews (below this email), we would like to invite the resubmission of a significantly-revised version that takes into account the reviewers' comments.

Thank you for submitting your work to PLOS Computational Biology. We appreciate the fully open-source and self-contained software framework for the computational evaluation of peripheral nerve stimulation, and acknowledge that it can potentially be highly useful for the community. Nevertheless, in the current state, several questions remain open. In particular, I would like to stress that more details on the optimization and the implementation of the parallelization with the MPI are needed (see comments of reviewer 1). In addition, the Discussion indicates that there are examples with detailed documentation online, which were (at the time of the review) not available on https://nrv.readthedocs.io/. Also on GitHub, only a very short example without documentation was present. Given that the usability of the framework is an important aspect of this work, we expect the authors to finish the online documentation for further consideration of the manuscript.

We cannot make any decision about publication until we have seen the revised manuscript and your response to the reviewers' comments. Your revised manuscript is also likely to be sent to reviewers for further evaluation.

Sincerely,

Bettina Christine Schwab, Ph.D.

Guest Editor

PLOS Computational Biology

Daniele Marinazzo

Section Editor

PLOS Computational Biology

Thank you for submitting your work to PLOS Computational Biology. We appreciate the fully open-source and self-contained software framework for the computational evaluation of peripheral nerve stimulation, and acknowledge that it can potentially be highly useful for the community. Nevertheless, in the current state, several questions remain open. In particular, I would like to stress that more details on the optimization and the implementation of the parallelization with the MPI are needed (see comments of reviewer 1). In addition, the Discussion indicates that there are examples with detailed documentation online, which were (at the time of the review) not available on https://nrv.readthedocs.io/. Also on GitHub, only a very short example without documentation was present. Given that the usability of the framework is an important aspect of this work, we expect the authors to finish the online documentation for further consideration of the manuscript.

Reviewer's Responses to Questions

**Comments to the Authors:**

Reviewer #1: Uploaded as attachment

Reviewer #2: The MS is a well written clear description of an open source model of peripheral nerve stimulation. The MS also includes a brief review and comparison to existing similar models.

As noted by the authors, the major issue for the model is the lack of ability to digest histological data for more realistic simulations. Another short coming is that all nerve are modelled as straight tubes, again missing the complexity of realistic nerves.

Reviewer #3: The review is in attachment

**Have the authors made all data and (if applicable) computational code underlying the findings in their manuscript fully available?**

Reviewer #1: **No: **Can't find any of the contents listed in "Supporting information", online documentation pages are incomplete (some say "currently writing it!" ... e.g. https://nrv.readthedocs.io/en/latest/quickstart/electrodes.html)

Reviewer #2: Yes

Reviewer #3: Yes

PLOS authors have the option to publish the peer review history of their article (what does this mean?). If published, this will include your full peer review and any attached files.

Reviewer #1: No

Reviewer #2: **Yes: **James Fallon

Reviewer #3: **Yes: **Mattia Stefano
---

## [Decision Letter · Decision Letter 1]

20 Jun 2024

Dear Dr Kolbl,

We are pleased to inform you that your manuscript 'NRV: An open framework for in silico evaluation of peripheral nerve electrical stimulation strategies' has been provisionally accepted for publication in PLOS Computational Biology.

Best regards,

Bettina Christine Schwab, Ph.D.

Guest Editor

PLOS Computational Biology

Daniele Marinazzo

Section Editor

PLOS Computational Biology

Reviewer 1 was not available anymore for the review process. The answers to their comments have been carefully checked by reviewer 3 and me, and all scientific concerns have been relieved. The manuscript still contains a large number of typos (e.g. line 108: optionnal, line 112: computationnal), which should ideally be corrected before the editorial process starts. I congratulate the authors on their nice work.

Reviewer's Responses to Questions

**Comments to the Authors:**

Reviewer #3: The review is uploaded as an attachment

**Have the authors made all data and (if applicable) computational code underlying the findings in their manuscript fully available?**

Reviewer #3: Yes

PLOS authors have the option to publish the peer review history of their article (what does this mean?). If published, this will include your full peer review and any attached files.

Reviewer #3: No

---

## [Editor Report · Acceptance letter]

4 Jul 2024

PCOMPBIOL-D-24-00070R1 

NRV: An open framework for in silico evaluation of peripheral nerve electrical stimulation strategies

Dear Dr Kolbl,

I am pleased to inform you that your manuscript has been formally accepted for publication in PLOS Computational Biology. Your manuscript is now with our production department and you will be notified of the publication date in due course.

With kind regards,

Zsofia Freund
